# TabImpute: Accurate and Fast Zero-Shot Missing-Data Imputation with a Pre-Trained Transformer

## Abstract

Missing data is a pervasive problem in tabular settings. Existing solutions range from simple averaging to complex generative adversarial networks, but due to each method's large variance in performance across real-world domains and time-consuming hyperparameter tuning, no default imputation method exists. Building on TabPFN, a recent tabular foundation model for supervised learning, we propose TabImpute, a pre-trained transformer that delivers accurate and fast zero-shot imputations requiring no fitting or hyperparameter tuning at inference-time. To train and evaluate TabImpute, we introduce (i) an entry-wise featurization for tabular settings, which enables a $100\times$ speedup over the previous TabPFN imputation method, (ii) a synthetic training data generation pipeline incorporating realistic missingness patterns, and (iii) MissBench, a comprehensive benchmark with 42 OpenML datasets and 13 new missingness patterns. MissBench spans domains such as medicine, finance, and engineering, showcasing TabImpute's robust performance compared to 12 established imputation methods.

## 1 Introduction

Missing data is ubiquitous across tabular datasets, affecting statisticians, economists, health officials, and businesses. For example, healthcare datasets may lack some recorded blood pressure measurements, or datasets merged from multiple sources may only share partial features. Regardless of the source, missing data must be imputed to numerical values before employing statistical or machine learning models. Imputation methods range from simple approaches (e.g., constant values and averages) to more sophisticated techniques like nearest-neighbor-based methods (Batista & Monard, 2003), matrix factorization approaches such as SoftImpute (Hastie et al., 2015), and random forest regression, notably the MissForest algorithm (Stekhoven & Bühlmann, 2011). However, each method in the literature is typically tailored for specific settings, with performance varying significantly across datasets, domains, and missingness patterns (Van Buuren, 2012; Jarrett et al., 2022; Agarwal et al., 2023; Ibrahim et al., 2005). Building on recent advances in tabular representation learning (Hollmann et al., 2023; Ye et al., 2025), we propose a pre-trained transformer specifically designed for the tabular missing-data problem that produces accurate and fast zero-shot imputations.

Rubin (1976) proposed analyzing missingness based on its relationship with the data-generating process to determine whether missingness biases downstream estimation. Rubin demonstrated that when missingness operates independently of the underlying data, the observed data distribution provides an unbiased foundation for estimation. This framework categorizes missingness into three classes: Missing Completely At Random (MCAR), Missing At Random (MAR), and Missing Not At Random (MNAR) (Van Buuren, 2012; Sportisse et al., 2020a;b). MCAR defines scenarios where missingness occurs uniformly and independently of all data values. MAR encompasses cases where missingness depends on observed variables that can be appropriately conditioned upon during analysis. MNAR describes situations where missingness depends on unobserved factors that cannot be easily conditioned on.

While this framework characterizes how missingness relates to the underlying data-generating process, it largely treats the missingness indicators for each variable as conditionally independent across entries. Recent work on structured missingness argues that, especially in large-scale multi-source

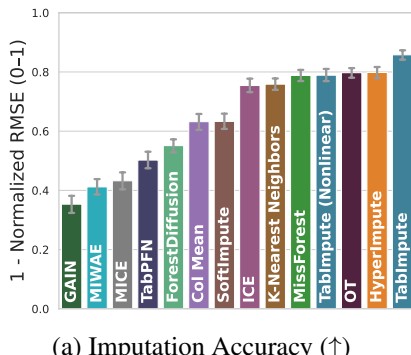 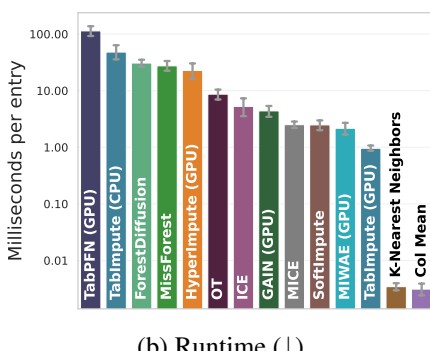

(a) Imputation Accuracy (↑)                    (b) Runtime (↓)

Figure 1: **Evaluation on real-world OpenML data: MissBench.** We compare TabImpute with 12 other popular methods on MissBench. In panel (a), we plot the imputation accuracy (defined as 1 - normalized RMSE), which is calculated for each method, normalized within a dataset, and averaged across datasets and 13 missingness patterns. Error bars indicate 95% confidence intervals. In panel (b), we compare the runtime per table entry. Any method not labeled (GPU) is tested on a CPU because that method is not implemented for GPUs. See Sec. 3 for our exact computing specifications and Sec. 4 for accuracy score methodology.

datasets, missingness itself can exhibit rich multivariate structure that is not captured by the standard MCAR/MAR/MNAR taxonomy. In particular, Mitra et al. (2023) introduce structured missingness as an umbrella term for mechanisms in which missingness follows systematic patterns. Jackson et al. (2023) provide a complementary characterization that allows dependencies among missingness indicators for different variables.

Previous work typically introduced new missingness patterns within these categories and proposed pattern-specific solutions. For instance, Agarwal et al. (2023) proposed a nearest neighbor-based matrix completion method specifically designed for a block-wise MNAR pattern. We instead develop a single method that performs well across diverse patterns and data domains by building on TabPFN, a popular tabular foundation model for supervised learning.

TabPFN is a pre-trained transformer model for supervised learning that performs well across a variety of domains without any fine-tuning (Hollmann et al., 2025). The team behind TabPFN created an imputation method in their `tabpfn-extensions` Python package by using the TabPFN model in iterative column-wise imputation (available on GitHub[1]). However, when evaluated on our new benchmark MissBench–consisting of 42 real-world OpenML datasets and 13 missingness patterns– TabPFN's imputation approach shows low accuracy and slow runtime. We improve on this approach by introducing a new entry-wise featurization (EWF), allowing parallel prediction of each missing value using TabPFN's model, denoted EWF-TabPFN. To further improve on EWF-TabPFN, we train a new underlying model specifically designed for tabular data imputation, TabImpute, to better fit this class of tasks, achieving the imputation accuracy shown in Fig. 1 and Tab. 1.

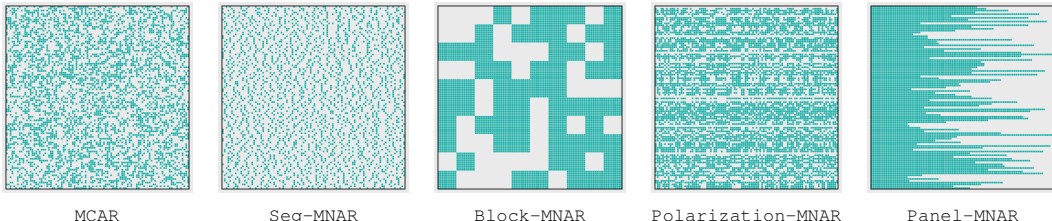

Figure 2: **Selection of synthetic missingness patterns implemented in MissBench.** Blue entries indicate observed values, and gray entries are unobserved.

---

[1] https://github.com/PriorLabs/tabpfn-extensions/blob/main/src/tabpfn_extensions/unsupervised/unsupervised.py

The main contributions of this work can be summarized as follows (also shown in Fig. 3):

- We propose a new state-of-the-art pre-trained transformer model, TabImpute, for missing data imputation by building on TabPFN and introducing a new entry-wise missing data featurization (see Sec. 3.1 for details).
- We develop a synthetic data generation pipeline to create training datasets with a comprehensive collection of missing values covering a wide range of MNAR patterns (see Sec. 3.2, App. A.2, and Tab. 9 for details).
- We demonstrate TabImpute's performance on a novel, comprehensive test bench, denoted MissBench, using 42 real-world OpenML datasets and 13 missingness patterns (see Sec. 4 for details). Our benchmark builds on previous work by testing more datasets and more missingness patterns: HyperImpute (Jarrett et al., 2022) tests on 13 UC Irvine (UCI) datasets (Kelly et al., 2024), a subset of OpenML, with 3 missingness patterns, and GAIN (Yoon et al., 2018) tests on 6 UCI datasets with 1 missingness pattern.

Our code and implementation details for all our contributions above can be accessed on GitHub.[2]

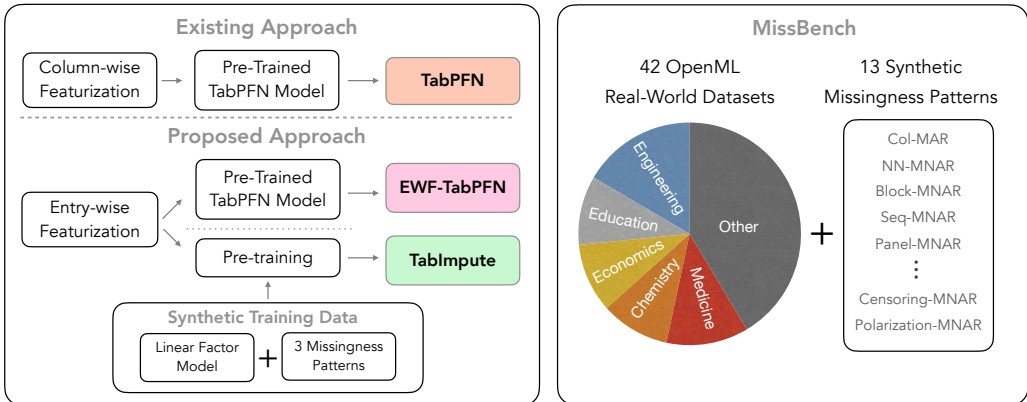

Figure 3: **Overview of our contributions.** The first row demonstrates TabPFN's imputation method, which performs iterative column-by-column imputation. We build on this by introducing an entry-wise featurization, as shown in the second row. We create a new synthetic data-generator for missingness data to train our model, TabImpute, shown in green (Sec. 3.2 and Sec. 3.3, respectively). We evaluate all the imputers on the comprehensive and rich set of OpenML datasets with many missingness patterns applied (Sec. 4).

### 1.1 PREVIOUS WORK

We build primarily on missing data imputation and tabular representation learning (TRL). While missing data imputation is well-studied with established theory dating back to the 1970s (Rubin, 1976), TRL using tabular foundation models is relatively new (Müller et al., 2022; Zhang et al., 2025; Hollmann et al., 2023; 2025). Below, we describe relevant work that we directly compare against or build upon.

**Imputation methods.** Given the widespread nature of missing data, numerous imputation techniques have been proposed. These include averages, linear models over columns (Efron, 1994), random forest models (Hindy et al., 2024), ensemble methods (Jarrett et al., 2022), nearest neighbor-based methods (Chin et al., 2025), and even using generative adversarial networks (Yoon et al., 2018). For fully numerical data, matrix completion methods like SoftImpute (Hastie et al., 2015) have also been employed.

HyperImpute (Jarrett et al., 2022) combines the power of multiple classical imputation methods through iterative imputation (IM). IM loops over each column, using other columns to predict missing values until convergence. HyperImpute optimizes the imputer at each iteration over candidate

---

[2]https://anonymous.4open.science/r/tabular-6F65/README.md

methods, including MICE (Royston & White, 2011), SoftImpute (Hastie et al., 2015), column mean, MissForest (Stekhoven & Bühlmann, 2011), and Optimal Transport (Muzellec et al., 2020). While achieving great imputation accuracy across diverse missingness patterns and supporting categorical variables, HyperImpute is designed for supervised learning settings and cannot handle matrices with entirely missing columns (common in causal inference and panel data (Agarwal et al., 2023)). Additionally, optimization requires significantly more time than base methods.

**Tabular representation learning.** Tabular representation learning focuses on building models that can generalize across diverse tabular domains. The pioneering work in this area is TabPFN, a tabular foundation model for supervised learning. TabPFN was proposed in Hollmann et al. (2023) and subsequently improved in Hollmann et al. (2025). Since TabPFN's introduction, numerous variants have emerged to address scalability and performance limitations, as well as further work aimed at clarifying its internal representations (Zhang et al., 2025; Ye et al., 2025). Recent advances include TabICL (Qu et al., 2025), a scalable foundation model that extends supervised learning capabilities to datasets with up to 500K samples through a novel two-stage architecture with column-then-row attention mechanisms. Other notable models include MITRA (Zhang & Robinson, 2025), a tabular foundation model pre-trained on purely synthetic data from a mix of random classifiers/regressors, and Toto (Cohen et al., 2025), which is optimized for time series forecasting on observability metrics. Additionally, CausalFM (Ma et al., 2025) enables Bayesian causal inference through structural causal model priors similar to TabPFN, while DO-PFN (Robertson et al., 2025) extends the PFN framework to estimate counterfactual distributions from observational data.

## 2 BACKGROUND ON PRIOR-DATA FITTED NETWORKS AND TABPFN

Prior-data Fitted Networks (PFNs) are a class of models that learn to approximate Bayesian inference for a given prior (Müller et al., 2022). Instead of fitting a new model from scratch, a PFN is an individual, large pre-trained Transformer to perform classification or regression in a single forward pass. This process, known as in-context learning (ICL), allows the model to make predictions using sequences of labeled examples provided directly in the input, without requiring any gradient updates (Dong et al., 2024). The entire prediction algorithm is contained in the weights of the network, which is trained once on millions of synthetically generated datasets sampled from the prior. At inference time, the trained PFN takes a real-world dataset, composed of training and test samples, as a set-valued input and returns a distribution over the output space. This output space is categorical for classification tasks and the real line for regression tasks.

**Posterior predictive modeling and synthetic prior fitting.** PFNs are rooted in Bayesian supervised learning, where the primary objective is to model the posterior predictive distribution (PPD) (MacKay, 1992; Seeger, 2004; Blei et al., 2017). Since computing the PPD is often intractable (MacKay, 1992), PFNs instead learn to approximate the PPD offline through a process called synthetic prior fitting Müller et al. (2022). This is achieved using a prior specified by a sampling scheme that first samples a data-generating mechanism, $\phi \sim p(\phi)$, and then samples a synthetic dataset, $D \sim p(D|\phi)$. This process is repeated to generate millions of diverse datasets for training. The network's parameters, $\theta$, are then optimized to predict held-out test samples ($D_{\text{test}} \subset D$) conditioned on the rest of the dataset ($D_{\text{train}} = D \setminus D_{\text{test}}$). The training objective is to minimize the negative log likelihood (NLL) loss on these held-out examples:

$$\mathcal{L}_{\text{NLL}}(\theta) = \mathbb{E}_{((x_{\text{test}}, y_{\text{test}}) \cup D_{\text{train}}) \sim p(D)}[-\log q_\theta(y_{\text{test}}|x_{\text{test}}, D_{\text{train}})].$$

This training process makes it explicit that PFNs are trained to emulate Bayesian inference by averaging over tasks drawn from a prior distribution over data-generating functions. Minimizing this loss ensures that the trained neural network, $q_\theta$, learns to approximate the true Bayesian PPD for the specified prior.

**TabPFN.** TabPFN is a PFN built specifically for tabular supervised learning (Hollmann et al., 2023). The model employs a novel two-way attention mechanism specifically designed for tabular data. Unlike standard transformers (Vaswani et al., 2023) that treat tabular data as sequential tokens, TabPFN assigns separate representations to each cell in the table. The architecture uses alternating attention patterns: each cell first attends to other features within its row (inter-feature attention), then attends to the same feature across all rows (inter-sample attention). This design ensures permutation

invariance for both samples and features while enabling efficient scaling to larger tables than those seen during training. TabPFN v2 (Hollmann et al., 2025) retains the core training paradigm of the original TabPFN while introducing several key enhancements that improve accuracy, runtime, and applicability. Going forward, when referring to TabPFN, we mean TabPFN v2 since it is the most up-to-date version.

## 3    TRAINING TABIMPUTE ON SYNTHETIC DATA

We develop a featurization for tabular missing data imputation that enables better utilization of TabPFN's architecture, create a synthetic data generation pipeline across diverse missingness patterns, and employ an adaptive training algorithm to balance performance across all patterns. Training used 8 H200 GPUs and an Intel Xeon Platinum 8592+ CPU over approximately 2 days, processing 1.9 million synthetic tables. Our model matches TabPFN's size and runs on CPU-only systems. Evaluation used 1 H200 GPU.

### 3.1    ENTRY-WISE FEATURIZATION AND ARCHITECTURE

We recast missing data imputation as supervised learning to leverage TabPFN's architecture and enable parallel GPU computation of missing entries. For each dataset (training point), let $X^*$ be the complete matrix with $m$ rows and $n$ columns, $\Omega$ be the set of missing entry indices, and $X$ be the matrix with induced missingness:

$$X_{ij} = \begin{cases} X_{ij}^* & \text{for} \quad (i,j) \in [m] \times [n] \setminus \Omega \\ \star & \text{otherwise.} \end{cases}$$

where $\star$ denotes a missing entry. Let $\Omega_{\text{obs}} = [m] \times [n] \setminus \Omega$. Our feature matrix construction adds $(i \oplus j \oplus X_{i,:} \oplus X_{:,j})$ for each entry $i, j \in [m] \times [n]$, where $X_{i,:}$ denotes the $i$-th row, $X_{:,j}$ the $j$-th column, and $\oplus$ concatenation. Each row's target is $y_{ij} = X_{ij}^*$. During pre-training, we train the model to predict target values for all $(i, j) \in \Omega$. This procedure creates a feature matrix of size $nm \times (n + m)$. This featurization captures all necessary information for each cell through its row and column context while enabling parallel computation of missing entries on GPUs. Although the input matrix size increases, parallelization gains outweigh this cost.

**Architecture.**    We use TabPFN's base architecture with one modification: removing the attention mask to allow input rows to attend to query rows. Since our input/query row sizes vary randomly with missingness patterns (unlike TabPFN's controlled synthetic generation), we remove the mask to enable parallel batch training. TabPFN's mask prevents train points from seeing test feature distributions, which is important in general supervised learning. However, our *test* set is created using data already available to the observed points, thus alleviating any data-leakage concerns.

### 3.2    SYNTHETIC TRAINING DATA GENERATION

We generate around 1.9 million matrices with missing values to train our model through a two-step process: first, generating underlying data, then introducing missingness patterns on top.

#### 3.2.1    DATA GENERATION WITH LINEAR FACTOR MODELS

We generate data using a simple linear factor model (LFMs) (Bai & Ng, 2002). LFMs are commonly used in matrix completion literature to prove error bounds for matrix completion algorithms (Koren et al., 2009; Candes & Recht, 2012). This family of models generates a data matrix $Y \in \mathbb{R}^{m \times n}$ by assuming the data lies on or near a low-dimensional subspace. The simplest case generates the data matrix $Y$ as the inner product of two lower-rank latent factor matrices, $U \in \mathbb{R}^{m \times k}$ and $V \in \mathbb{R}^{n \times k}$, where $k \ll n, m$ is the rank:

$$Y = UV^T.$$

To generate diverse datasets, the latent vectors (rows of $U$ and $V$) are sampled from a variety of distributions, including Gaussian, Laplace, Student's t, spike-and-slab (a mixture of a Dirac delta at zero and a Gaussian), and Dirichlet.

When training, we experimented with several classes of data-generating processes (DGP), including matrices from nonlinear factor models and structural causal models (SCM) similar to the ones used in TabPFN. We found that training on SCM's proved too computationally expensive, but were able to train a model on nonlinear factor models. We found that the model trained on linear factor models performed the best, as shown in Tab. 8 in App. A.3. We leave it as future work to explore other DGPs which might further enhance the underlying model.

### 3.2.2 Missingness Patterns for Training and Evaluation

After generating a complete data matrix $X^* \in \mathbb{R}^{m \times n}$, we introduce missingness by applying a masking matrix $M \in \{0,1\}^{m \times n}$, where the entry value $M_{ij} = 1$ if and only if $(i,j) \in \Omega_{\text{obs}}$. To ensure TabImpute is robust and generalizable to the variety of ways data can be missing in real-world scenarios, we pre-train on a comprehensive stock of synthetic datasets with several missingness patterns. For convenience, we define $p_{ij} = \mathbb{P}(M_{ij} = 1)$, the propensity of each entry in $X^*$.

We include 13 different missingness patterns: 1 MCAR, 1 MAR pattern, and 11 MNAR patterns. For examples of these, see Fig. 2. MNAR patterns often stump standard imputation methods and yet are extremely common in the real world.

**MCAR:** MCAR missingness means the probability of an entry being missing, defined as its propensity, is constant across entries and independent from any other randomness. The missingness indicators $M_{ij}$ are drawn i.i.d. from a Bernoulli distribution $M_{ij} \sim \text{Bern}(p)$ across all $(i,j) \in [m] \times [n]$ for some constant $p \in (0,1)$. This is the simplest form of missingness, but is unrealistic (Van Buuren, 2012).

**MAR:** For MAR missingness, the probability of an entry being missing depends only on the observed values $X$. In other words, the randomness in MAR can be explained by conditioning on observed factors. Additionally, every entry has a positive probability of being observed (i.e., $p_{ij} > 0$). We simulate MAR through column-wise MAR, denoted Col-MAR: we choose several columns as predictor columns and use those values to mask entries in other columns. This is similar to the MAR approach taken in Jarrett et al. (2022).

**MNAR Patterns:** For the most complex missingness class, MNAR, the probability of an entry being missing can depend on unobserved factors. Note that MNAR patterns are significantly more difficult to handle systematically, often requiring specialized methods for a specific kind of MNAR pattern (Van Buuren, 2012). Due to the flexibility of our entry-wise featurization, TabImpute can produce imputations for these highly complex scenarios, even when columns are completely missing, such as in panel-data missingness patterns. HyperImpute was tested on two MNAR patterns in the Appendix of Jarrett et al. (2022), one where values are further masked after an MAR pattern and another where values outside a certain range are masked. We build on this work by testing on 11 MNAR patterns (see App. A.2 and Tab. 9 for details). We implement a range of MNAR patterns to simulate plausible real-world scenarios. For example, we utilize the expressiveness of neural networks to create random propensity functions MNAR patterns, use bandit algorithms to induce column-adaptive missing patterns, simulate panel data missingness where some features are removed later, censoring where sensor readings fall outside a detectable range, and survey data artifacts like respondent polarization and skip-logic.

### 3.3 Training TabImpute

We train our model to predict unobserved values under several missingness patterns simultaneously. For the final TabImpute model, we trained only on MCAR missingness because we found our model generalized to the other patterns well without including them explicitly in training. In fact, we found that including other missingness patterns degraded overall performance. We discuss these results further below. We use the prior-data fitted negative log likelihood (NLL) loss proposed in Müller et al. (2022). Like other PFNs with continuous numerical output, we use the Riemann distribution output also proposed in Müller et al. (2022). Since we can generate an unlimited amount of synthetic data, we do not reuse any synthetic data and only do one gradient pass per batch of datasets. This allows our model to learn the underlying data-generating process and missingness

mechanisms without risk of memorization. We use a learning rate of $0.0001$, a batch size of 16, and train on around 1.9 million synthetic datasets.

**Remark:** We had initially attempted to train the model sequentially on one missingness pattern at a time, but found that the network always experienced *catastrophic forgetting* (McCloskey & Cohen, 1989) irrespective of learning rate (i.e., it forgot how to handle the previous missingness patterns). After this, we attempted to mix several missingness types together: `MCAR`, `MAR`, and `Self-Masking-MNAR`. However, we found that the model's performance degraded overall because it focused only `Self-Masking-MNAR`. We attempted several mitigation techniques including pattern weight schedules and GradNorm Chen et al. (2018), a gradient normalization technique for multi-task learning, but found them unsuccessful in improving performance.

## 4    RESULTS ON OPENML DATASETS: MISSBENCH

To evaluate TabImpute against other methods, we introduce MissBench: a missing-data imputation benchmark using $42$ OpenML (Vanschoren et al., 2013) tabular datasets with 13 synthetic missingness patterns. For every dataset and missingness pattern, we test each method's ability to impute masked values. The $42$ OpenML datasets span domains such as medicine, engineering, and education. The missingness patterns include 1 `MCAR` pattern, 1 `MAR` pattern, and 11 `MNAR` patterns. We provide details for the `MNAR` patterns in App. A.2.

We test a suite of imputation methods on MissBench: column-mean imputation (Hawthorne & Elliott, 2005), SoftImpute (Hastie et al., 2015), MissForest (Stekhoven & Bühlmann, 2011), iterative chained estimators (ICE/MICE) (van Buuren & Groothuis-Oudshoorn, 2011; Royston & White, 2011), GAIN (Yoon et al., 2018), MIWAE (Mattei & Frellsen, 2019), an optimal transport-based method (Muzellec et al., 2020), and TabPFN's imputation method (see Sec. 1). We also test imputing categorical variables via one-hot encoding in App. A.1 and report $R^2$ values in Tab. 7.

**Imputation Accuracy.** To ensure a fair comparison across datasets with different scales and inherent difficulties, we report a normalized accuracy score. In particular, for each dataset and missingness pattern, we first calculate the standard Root Mean Squared Error (RMSE) for every imputation method as $\left(\frac{1}{|\Omega|}\sum_{(i,j)\in\Omega}\left(X_{ij}^{\text{true}}-X_{ij}^{\text{imputed}}\right)^2\right)^{1/2}$, where $\Omega$ denotes the set of missing entries. We then perform a min-max normalization on these RMSE scores across all methods for that specific task:

$$\text{Normalized RMSE} = \frac{\text{RMSE}_{\text{method}} - \min(\text{RMSE}_{\text{all methods}})}{\max(\text{RMSE}_{\text{all methods}}) - \min(\text{RMSE}_{\text{all methods}})}$$

This normalization maps the best-performing method to 0 and the worst to 1. Finally, we define *Imputation Accuracy* as $1 - $ Normalized RMSE, where higher values indicate better performance.

The final scores we report represent imputation accuracy averaged across all $42$ datasets and 13 missingness patterns. Dataset sizes range from $50 \times 5$ to $170 \times 55$. We evaluate only on datasets with numerical values without pre-existing missingness before applying synthetic patterns. Specific datasets are listed in Tab. 10 (App. A.3).

Tab. 1 presents results for each missingness pattern as well as overall performance. TabImpute achieves the best overall performance and for nearly all individual patterns. For completeness, we list the performance of methods not shown in the table in Tab. 6 and non-normalized RMSE values in Tab. 11. TabImpute performs best under high missingness conditions (Fig. 4, App. A.1), which is expected since it leverages generative pre-training rather than relying solely on available dataset information like discriminative methods.

**Multiple imputation (MI) Metrics.** In multiple imputation, a dataset is imputed multiple times independently. These datasets can then be used to provide better confidence intervals of downstream quantities Van Buuren (2012). Given two imputation methods that have the same imputation accuracy, one would prefer a method with higher uncertainty to not provide false precision on imputed values Van Buuren (2012); Li et al. (2015); Jolicoeur-Martineau et al. (2024). We assess performance for multiple imputation by repeating imputation 5 times per dataset and calculating the median absolute deviation (MAD), minimimum RMSE, and average RMSE, shown in Tab. 2. While

Table 1: Imputation Accuracy ± Standard Error by Missingness Pattern.
We train on the pattern above the dashed line. The number of samples used to calculate standard error are the number of datasets: 42.

| Pattern | TabImpute | EWF-TabPFN | HyperImpute | MissForest |
|---|---|---|---|---|
| MCAR | **0.882 ± 0.021** | 0.817 ± 0.024 | 0.804 ± 0.031 | 0.867 ± 0.022 |
| NN-MNAR | **0.900 ± 0.019** | 0.876 ± 0.020 | 0.760 ± 0.036 | 0.818 ± 0.029 |
| Self-Masking-MNAR | **0.703 ± 0.038** | 0.682 ± 0.042 | 0.689 ± 0.039 | 0.638 ± 0.040 |
| Col-MAR | **0.877 ± 0.030** | 0.855 ± 0.027 | 0.814 ± 0.040 | 0.773 ± 0.039 |
| Block-MNAR | 0.890 ± 0.026 | **0.905 ± 0.026** | 0.873 ± 0.027 | 0.860 ± 0.023 |
| Seq-MNAR | **0.920 ± 0.014** | 0.905 ± 0.015 | 0.864 ± 0.030 | 0.829 ± 0.031 |
| Panel-MNAR | **0.915 ± 0.025** | 0.711 ± 0.048 | 0.865 ± 0.033 | 0.912 ± 0.020 |
| Polarization-MNAR | 0.804 ± 0.024 | **0.879 ± 0.021** | 0.622 ± 0.039 | 0.567 ± 0.033 |
| Soft-Polarization-MNAR | 0.759 ± 0.045 | **0.864 ± 0.024** | 0.711 ± 0.034 | 0.667 ± 0.041 |
| Latent-Factor-MNAR | **0.891 ± 0.021** | 0.883 ± 0.018 | 0.775 ± 0.036 | 0.834 ± 0.023 |
| Cluster-MNAR | **0.903 ± 0.014** | 0.871 ± 0.021 | 0.839 ± 0.028 | 0.830 ± 0.021 |
| Two-Phase-MNAR | **0.886 ± 0.026** | 0.851 ± 0.029 | 0.865 ± 0.031 | 0.873 ± 0.020 |
| Censoring-MNAR | 0.710 ± 0.032 | 0.593 ± 0.045 | **0.786 ± 0.041** | 0.671 ± 0.038 |
| Overall | **0.849 ± 0.008** | 0.823 ± 0.009 | 0.790 ± 0.010 | 0.780 ± 0.009 |

most imputation methods require another full run in order to create another imputation, TabImpute and other PFN-based methods output a distribution. This distribution is then sampled 5 times to create 5 different imputations, following the methodology of ForestDiffusion Jolicoeur-Martineau et al. (2024). Thus, multiple imputation for TabImpute uses the same runtime as a single imputation.

TabImpute provides better diversity than methods of comparable imputation accuracy (MissForest and HyperImpute). The poor precision of MissForest and HyperImpute is shown through their worse minimum RMSE values. Note that in Tab. 2, TabImpute has a higher avergage RMSE value then MissForest and HyperImpute. This is because it has a higher diversity in its estimate (the samples have a larger spread). When comparing just the median estimate, though, TabImpute has a better accuracy, as shown in the first row in Tab. 1. Thus, TabImpute provides practitioners with the best of both worlds: If they require multiple imputation for uncertainty estimates, TabImpute provides diverse estimates, and if they require a single good imputation, TabImpute provides the most accurate estimate.

Table 2: Multiple Imputation Metrics: Dataset-level MAD and RMSE Metrics ± Standard Error per Imputed Entry Across 5 Samples (MCAR, p=0.4).

| Method | MAD ($\uparrow$) | Min RMSE ($\downarrow$) | Avg RMSE ($\downarrow$) |
|---|---|---|---|
| MICE | **0.359 ± 0.002** | **0.264 ± 0.003** | 0.859 ± 0.005 |
| MIWAE | 0.267 ± 0.001 | 0.385 ± 0.005 | 0.927 ± 0.005 |
| GAIN | 0.200 ± 0.001 | 0.361 ± 0.004 | 0.789 ± 0.005 |
| TabImpute | 0.199 ± 0.002 | 0.337 ± 0.004 | 0.722 ± 0.006 |
| ForestDiffusion | 0.181 ± 0.001 | 0.291 ± 0.004 | 0.739 ± 0.005 |
| OT | 0.049 ± 0.000 | 0.440 ± 0.005 | 0.547 ± 0.005 |
| HyperImpute | 0.037 ± 0.001 | 0.420 ± 0.004 | 0.551 ± 0.005 |
| MissForest | 0.036 ± 0.000 | 0.434 ± 0.005 | **0.532 ± 0.005** |
| SoftImpute | 0.001 ± 0.000 | 0.577 ± 0.005 | 0.649 ± 0.005 |
| Col Mean | 0.000 ± 0.000 | 0.813 ± 0.005 | 0.813 ± 0.005 |
| ICE | 0.000 ± 0.000 | 0.606 ± 0.005 | 0.606 ± 0.005 |

**Specialized TabImpute Models.** After training TabImpute on the MCAR pattern, we attempted to improve it by mixing in Col-MAR and Self-Masking-MNAR in the pre-training process. However, we found that the model consistently collapsed to only learning Self-Masking-MNAR, degrading performance on other patterns. This new model, though, performed far better on Self-Masking-MNAR than any other method, demonstrating the power of this architecture and

pre-training to learn any pattern. We show the relative performance of three models trained on `MCAR`, `Col-MAR`, and `Self-Masking-MNAR`, respectively, in Tab. 3.

The model trained on `Self-Masking-MNAR` performs much better than the other models on this pattern. However, it does not generalize well to other patterns because the masks generated by the `Self-Masking-MNAR` pattern do not cover the types of masks generated by other patterns. On the other hand, when generating `MCAR` patterns millions of times, a small portion of these masks will look like `Self-Masking-MNAR` masks. Interestingly, the `MCAR`-trained model performed slightly better on `Col-MAR` patterns than the model trained on `Col-MAR` exclusively. One explanation for this is that the masks generated under `MAR` pattern are covered well by `MCAR` masks and `MCAR` masks produce a better underlying model. Note that the imputation accuracy scores are different for Tab. 1 and Tab. 3 because the `MAR` and `Self-Masking-MNAR` models are not included in the normalization in Tab. 1. However, the order of the imputation accuracy remains consistent.

**Remark:** We implemented an ensembling approach which weighted specialized model outputs by how well they predicted the observed values within a matrix, but found this approach to always favor the `MCAR`-trained model. We leave it for future work to determine a better method for ensembling or routing the specialized model predictions to better take advantage of specialized models.

Table 3: Imputation Accuracy ± Standard Error by Missingness Pattern for Specialized Models

| Pattern | TabImpute$_{\text{MCAR}}$ | TabImpute$_{\text{Col-MAR}}$ | TabImpute$_{\text{Self-Masking-MNAR}}$ |
|---|---|---|---|
| MCAR | **0.938 ± 0.028** | 0.832 ± 0.024 | 0.020 ± 0.020 |
| Self-Masking-MNAR | 0.463 ± 0.067 | 0.329 ± 0.065 | **0.624 ± 0.072** |
| Col-MAR | **0.899 ± 0.042** | 0.811 ± 0.050 | 0.081 ± 0.041 |
| Overall | **0.766 ± 0.034** | 0.656 ± 0.035 | 0.243 ± 0.037 |

## 5 CONCLUSION & FUTURE WORK

In this paper, we present a state-of-the-art pre-trained transformer for the tabular missing data problem. We build on recent work in tabular representation learning by adapting TabPFN's architecture and training pipeline for the missing data setting. Even though we train purely on synthetic data, we are able to impute entries accurately on real-world OpenML data for a comprehensive set of missingness patterns, thus showcasing our model's ability to generalize to unseen domains. We open-source not only our model architecture and weights, but also our training and evaluation code (available at `https://anonymous.4open.science/r/tabular-6F65/README.md`). We hope this will facilitate others to validate, utilize, and build upon our work.

Since we use the same architecture as TabPFN, we also suffer from the quadratic time complexity of attention along both rows and columns. Due to entry-wise featurization, this complexity is squared again along the row axis. While TabImpute proved to be fast on relatively small tables in MissBench, we expect scalability to be a concern on larger table sizes. Additionally, TabImpute is very fast when using a GPU, but is slower on a CPU similar to TabPFN. This can be partly alleviated by imputing tables in chunks, but we leave this for future work.

Note that since we utilize a base architecture similar to TabPFN, any further improvements to TabPFN's architecture can be immediately ported to TabImpute. For example, Zeng et al. (2025) and Qu et al. (2025) propose different attention mechanisms to speed up TabPFN-like architectures for tabular data. Finally, since we use a PFN architecture, we output a distribution for each missing entry and can sample from this distribution for multiple imputation (Rubin, 2018). In future work, we plan on (i) exploring training further on more complex missingness patterns and data-generating processes, (ii) enhancing our method to impute categorical data better, (iii) extending our evaluation set to causal inference settings, which can be modeled as missing-data problems (Agarwal et al., 2023), (iv) improving the architecture to scale to larger datasets, and (v) utilizing TabImpute further for multiple imputation datasets.

**Reproducibility Statement.** We open-source our training code, evaluation code, model architecture and weights, and synthetic data-generation pipeline (available at `https://anonymous.4open.science/r/tabular-6F65/`). While we utilize 8 H200 GPUs to train our model, our model should be able to fit easily on GPUs with far less memory because our model only has around 22 million parameters. All of the datasets we use for evaluation are public and available in OpenML, making it straightforward to reproduce our results from our pre-trained model.

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

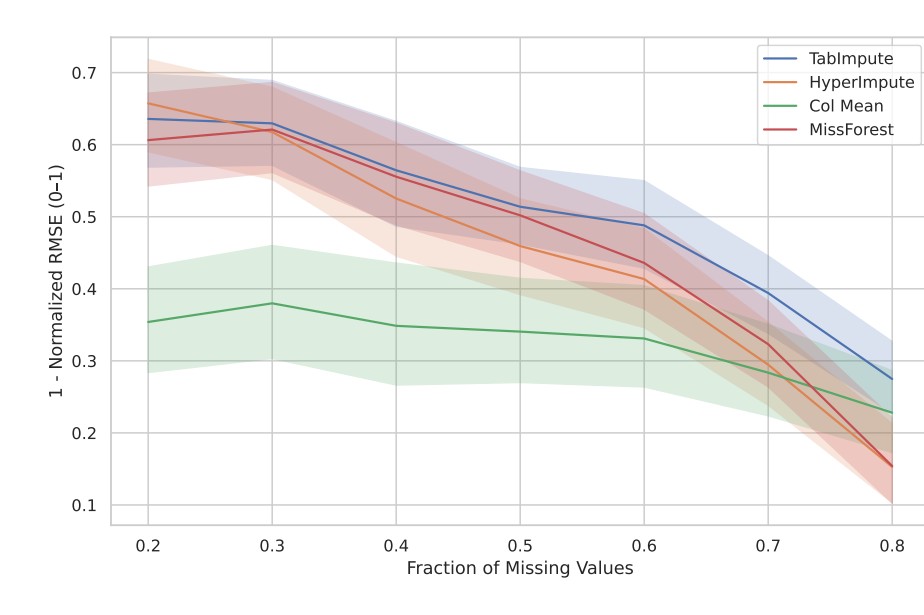

Figure 4: **Imputation Accuracy vs. probability of missingness for `MCAR`.** TabImpute performs the best when missingness is higher because it is a generative model that fits to the data in-context.

Han-Jia Ye, Si-Yang Liu, and Wei-Lun Chao. A closer look at tabpfn v2: Strength, limitation, and extension. *arXiv preprint arXiv:2502.17361*, 2025.

Jinsung Yoon, James Jordon, and Mihaela van der Schaar. Gain: Missing data imputation using generative adversarial nets. In *International Conference on Machine Learning (ICML)*, 2018.

Yuchen Zeng, Tuan Dinh, Wonjun Kang, and Andreas C Mueller. Tabflex: Scaling tabular learning to millions with linear attention. *arXiv preprint arXiv:2506.05584*, 2025.

Qiong Zhang, Yan Shuo Tan, Qinglong Tian, and Pengfei Li. Tabpfn: One model to rule them all?, 2025. URL `https://arxiv.org/abs/2505.20003`.

Xiyuan Zhang and Danielle Maddix Robinson. Mitra: Mixed synthetic priors for enhancing tabular foundation models, Jul 2025. URL `https://www.amazon.science/blog/mitra-mixed-synthetic-priors-for-enhancing-tabular-foundation-models`.

## A  APPENDIX

Here we present the rest of the missingness patterns we tested on, tables with further results, the methods we tested against, and the OpenML datasets we evaluated on.

### A.1  ADDITIONAL TESTS

Next, we discuss when TabImpute does well and when HyperImpute and other methods do well. We found that an important factor in determining performance was the level of missingness. The probability of missingness can only be controlled precisely for `MCAR`. Thus, we show in Fig. 4 the performance of the top methods as we increase the missingness level.

**Support for categorical variables.** We support imputing categorical variables via one-hot encodings: First, we convert each categorical column into several one-hot encoding columns. Then, we impute missing entries within this now purely numerical matrix. We then choose the class with the highest score as the categorical imputation for that missing entry. Note that we could also perform

a softmax operation over the imputed scores within the one-hot encoded columns to output probabilities over classes. With this method, TabImpute achieves AUC close to that of MissForest and HyperImpute when tested on `MCAR` missingness with $p = 0.4$, as shown in Tab. 4. We leave it for future work to incorporate categorical features into the training procedure, which would likely improve TabImpute's performance.

## A.2 DETAILS FOR MNAR MISSINGNESS PATTERNS

This section provides the mathematical and implementation details for each of the simulated `MNAR` missingness mechanisms that we implement and test. For each pattern, we define the mechanism by which the missingness mask $M$ is generated, where $M_{ij} = 1$ if the value $X_{ij}$ is observed and $M_{ij} = 0$ otherwise.

### A.2.1 DETAILS FOR NN-MNAR

**Description**  This pattern simulates a scenario where the propensity $p_{ij}$ depends on the underlying matrix values $X^*$ in an arbitrary manner. We achieve a comprehensive coverage of `MNAR` patterns by leveraging the expressiveness of neural networks.

**Methodology**  One general form of `MNAR` can be described as follows: for all $i$ and $j$, there exists some function $f_{ij}$ on the true (hence unobserved) matrix $X^*$ such that the propensity depends on $X^*$ as follows: $p_{ij}(X^*) = \mathbb{P}(M_{ij} = 1|X^*) = f_{ij}(X^*)$. By leveraging the expressiveness of neural networks, we propose a neural-net-based `MAR` pattern generator (`NN-MNAR`) that is designed to approximate arbitrary propensities characterized by functions $f_{ij}$.

**Implementation Details**  For fixed indices $i$ and $j$, `NN-MNAR` constructs the propensity $p_{ij}$ in a two-step procedure. First, we randomly collect a subset of values from the matrix $X^*$ and flatten them as a vector, say $X^*(i,j)$; we do this by first randomly generating a neighborhood $\mathbf{N}_{ij} \subset [m] \times [n]$, then the entries in the neighborhood $X^*_{st}, (s,t) \in \mathbf{N}_{ij}$ constitute the entries of the vector $X^*(i,j)$. Second, a neural-net function $g_{ij} : \mathbb{R}^{|\mathbf{N}_{ij}|} \to [0,1]$ is constructed by randomly initializing the number of layers, depth, weight, and bias. At each training step, the random neighborhood $\mathbf{N}_{ij}$ and the random neural-net $g_{ij}$ collectively defines the propensity $p_{ij} = g_{ij}(X^*(i,j))$ from which `MNAR` missingness patterns are generated $M_{ij} \sim \text{Bern}(p_{ij})$.

### A.2.2 DETAILS ON SEQ-MNAR

**Description**  This pattern simulates a scenario where masking matrix values $M_{ij} \in \{0,1\}$ for each column $j$ are adaptively chosen depending on the information up to column $j-1$ (i.e., regard columns as time). Specifically, we employ variants of bandit algorithms Lattimore & Szepesvári (2020) while regarding the binary masking matrix values as the two arms. Such patterns commonly arise in sequential experiments Ghosh et al. (2024).

**Methodology**  The true matrix $X^*$ is transformed to constitute the reward. For each designated column $j$, one of the following bandit algorithm utilizes the all the information of $X^*$ and $M$ up to column $j-1$ and chooses one of the two arms $\{0,1\}$ via one of the following algorithms: $\varepsilon$-greedy, upper-confidence bound (UCB), Thompson-sampling (Thompson, 1933) or gradient bandit.

**Implementation Details**  We generate exogenous Gaussian noise and add it to the true matrix $X^*$ and regard $X^*$ as the reward for arm 0 and its noisy version as the reward for arm 1. Then, starting from the first column with multiple rows as multiple agents, we randomly initiate (with random configurations) one of the four algorithms Lattimore & Szepesvári (2020): $\varepsilon$-greedy, Upper Confidence Bound (UCB) (Auer et al., 2002), Thompson sampling with random configurations (Thompson, 1933). Further, we have the option to randomly mix pooling techniques Ghosh et al. (2024) on top of any of the four algorithms.

### A.2.3 SELF-MASKING-MNAR

**Description:**  This pattern simulates a scenario where the probability of a value being missing is a direct function of the value itself. This pattern, commonly referred to as "self-masked `MNAR`",

Table 4: Area Under the Curve (AUC) Performance on Categorical Columns with MCAR ($p = 0.4$) missingness.

| OpenML Dataset | HyperImpute | MissForest | TabImpute | Mode |
|---|---|---|---|---|
| ChonicKidneyDisease | 0.602 | 0.582 | 0.525 | 0.500 |
| Dog Breeds Ranked | 0.566 | 0.579 | 0.539 | 0.500 |
| HappinessRank 2015 | 0.486 | 0.466 | 0.677 | 0.500 |
| MY DB | 0.480 | 0.514 | 0.521 | 0.500 |
| Online Sales | 0.790 | 0.809 | 0.869 | 0.500 |
| Parkinson Dataset | 0.665 | 0.640 | 0.523 | 0.500 |
| acute-inflammations | 0.761 | 0.756 | 0.723 | 0.500 |
| aids | 0.511 | 0.573 | 0.509 | 0.500 |
| analcatdata creditscore | 0.678 | 0.672 | 0.491 | 0.500 |
| analcatdata cyyoung8092 | 0.683 | 0.649 | 0.611 | 0.500 |
| analcatdata cyyoung9302 | 0.713 | 0.700 | 0.629 | 0.500 |
| analcatdata impeach | 0.751 | 0.749 | 0.706 | 0.500 |
| analcatdata ncaa | 0.549 | 0.547 | 0.500 | 0.500 |
| analcatdata wildcat | 0.749 | 0.682 | 0.674 | 0.500 |
| auto price | 0.749 | 0.802 | 0.761 | 0.500 |
| backache | 0.529 | 0.516 | 0.502 | 0.500 |
| blogger | 0.558 | 0.573 | 0.499 | 0.500 |
| caesarian-section | 0.534 | 0.499 | 0.518 | 0.500 |
| cloud | 0.436 | 0.409 | 0.440 | 0.500 |
| cm1 req | 0.677 | 0.658 | 0.602 | 0.500 |
| cocomo numeric | 0.629 | 0.618 | 0.655 | 0.500 |
| conference attendance | 0.500 | 0.512 | 0.499 | 0.500 |
| corral | 0.553 | 0.558 | 0.575 | 0.500 |
| cpu | 0.694 | 0.675 | 0.546 | 0.500 |
| fl2000 | 0.608 | 0.468 | 0.533 | 0.500 |
| flags | 0.584 | 0.583 | 0.517 | 0.500 |
| fruitfly | 0.602 | 0.604 | 0.593 | 0.500 |
| grub-damage | 0.620 | 0.584 | 0.541 | 0.500 |
| hutsof99 logis | 0.569 | 0.568 | 0.583 | 0.500 |
| iris | 0.885 | 0.885 | 0.827 | 0.500 |
| kidney | 0.599 | 0.460 | 0.498 | 0.500 |
| lowbwt | 0.587 | 0.582 | 0.547 | 0.500 |
| lung | 0.558 | 0.483 | 0.496 | 0.500 |
| lungcancer GSE31210 | 0.647 | 0.527 | 0.567 | 0.500 |
| lymph | 0.641 | 0.594 | 0.547 | 0.500 |
| molecular-biology promoters | 0.508 | 0.505 | 0.504 | 0.500 |
| mux6 | 0.507 | 0.481 | 0.502 | 0.500 |
| nadeem | 0.594 | 0.521 | 0.542 | 0.500 |
| nasa numeric | 0.596 | 0.594 | 0.561 | 0.500 |
| postoperative-patient-data | 0.468 | 0.494 | 0.496 | 0.500 |
| prnn crabs | 0.840 | 0.687 | 0.737 | 0.500 |
| prnn viruses | 0.563 | 0.588 | 0.551 | 0.500 |
| qualitative-bankruptcy | 0.652 | 0.694 | 0.645 | 0.500 |
| servo | 0.528 | 0.516 | 0.480 | 0.500 |
| sleuth case1202 | 0.544 | 0.590 | 0.521 | 0.500 |
| sleuth case2002 | 0.609 | 0.564 | 0.526 | 0.500 |
| sleuth ex2015 | 0.589 | 0.599 | 0.733 | 0.500 |
| sleuth ex2016 | 0.562 | 0.588 | 0.532 | 0.500 |
| tae | 0.544 | 0.592 | 0.533 | 0.500 |
| teachingAssistant | 0.533 | 0.528 | 0.499 | 0.500 |
| veteran | 0.520 | 0.520 | 0.502 | 0.500 |
| white-clover | 0.626 | 0.598 | 0.622 | 0.500 |
| zoo | 0.754 | 0.710 | 0.764 | 0.500 |
| Overall | $0.609 \pm 0.097$ | $0.593 \pm 0.097$ | $0.577 \pm 0.096$ | $0.500 \pm 0.000$ |

is widely utilized in the missing-data literature to model unbalanced class problems (Mohan, 2018; Sportisse et al., 2020b). A classic example occurs in income surveys, where individuals at the extremes of the income distribution, both very high and very low earners, are often less likely to disclose their income (Moore & Welniak, 2000). Similar patterns appear in other domains, such as substance use reporting or medical contexts where patients with severe symptoms might be less able to complete follow-up assessments (Ibrahim & Molenberghs, 2009).

Within the structured missingness taxonomy proposed by (Jackson et al., 2023), this mechanism is classified as **MNAR-UP** (Unstructured, Probabilistic). It is "Unstructured" because the missingness of an entry $(i, j)$ is conditionally independent of the missingness status of other entries, and "Probabilistic" as it is determined by a probability function rather than a deterministic rule.

**Methodology:** For each designated target column $j$, the probability of an entry $(i, j)$ being missing is determined by a logistic function of its value. The relationship is defined as $\mathbb{P}(M_{ij} = 0 | X_{ij}^*) = \sigma(\alpha \cdot X_{ij}^* + \beta_0)$ where $\sigma(z) = (1 + e^{-z})^{-1}$ is the sigmoid function.

**Implementation Details:** A random coefficient $\alpha$ is chosen from the set $\{-2, -1, 1, 2\}$ to introduce variability in the direction and magnitude of the value's effect on its missingness probability. The bias term $\beta_0$ is calibrated to achieve a target missingness proportion, $p$.

### A.2.4 CENSORING-MNAR

**Description:** This pattern models missingness arising from the physical limitations of measurement instruments, where values falling below a lower Limit of Detection or above an upper Limit of Quantification are not recorded. This results in left-censored or right-censored data, a common obstacle in environmental science, epidemiology, and biomedical research (Helsel, 2011; Chen et al., 2013). In fields like metabolomics and proteomics, this type of missingness is explicitly recognized as left-censored MNAR (Karpievitch et al., 2009; Lazar et al., 2016), as the probability of being missing is directly determined by the concentration falling below the detection threshold. Failing to appropriately account for this, or using simplistic substitution methods like replacing with zero, can introduce substantial bias. Modeling `Censoring-MNAR` is therefore essential for evaluating imputation methods on datasets generated by instrumentation with inherent sensitivity constraints.

`Censoring-MNAR` maps to the **MNAR-UD** (Unstructured, Deterministic) category in the structured missingness framework. It is "Unstructured" as the missingness of an entry is independent of other missingness indicators, and "Deterministic" because any value falling beyond the censoring threshold is missing with certainty.

**Methodology:** For each column $j$, a censoring direction (left or right) is chosen with equal probability. A cutoff value is determined based on a specified quantile, $q_{censor}$, of the set of currently observed (non-missing) values in that column. Let $X_{:,j}^*$ denote the set of observed values in column $j$, i.e., $X_{:,j}^* = \{X_{ij} \mid M_{ij} = 1\}$.

- **Left-Censoring:** All values in column $j$ that are less than the $q_{censor}$-th quantile of the observed values in that same column are set to missing. The threshold is a single scalar value calculated from the column's observed data.

$$M_{ij} = 0 \quad \text{if} \quad X_{ij} < \text{quantile}(X_{:,j}^*, q_{censor})$$

- **Right-Censoring:** All values in column $j$ that are greater than the $(1 - q_{censor})$-th quantile of the observed values in that column are set to missing. The threshold is a single scalar value.

$$M_{ij} = 0 \quad \text{if} \quad X_{ij} > \text{quantile}(X_{:,j}^*, 1 - q_{censor})$$

**Implementation Details:** The choice between left- and right-censoring is made randomly for each column with a probability of 0.5 for each. We introduce a hyperparameter $q_{censor}$ for the censoring quantile that controls the fraction of data to be censored from either tail of the distribution. For our evaluation, we use $q_{censor} = 0.25$.

### A.2.5 PANEL-MNAR

**Description:** This pattern simulates participant attrition (dropout) in longitudinal and panel data studies. This creates missingness where the dropout of a subject at a specific time point means all subsequent data for that subject is unobserved. Attrition is an inherent feature of long-term studies, including clinical trials and cohort studies (Little & Rubin, 2019; Van Buuren, 2012). The primary concern is "attrition bias," which arises when dropout is systematic—meaning those who leave the study differ significantly from those who remain (Hausman & Wise, 1979). This often constitutes an MNAR mechanism, particularly in clinical settings where patients might withdraw due to adverse effects or perceived lack of treatment efficacy, directly linking the dropout to the (unobserved) future outcomes (Ibrahim & Molenberghs, 2009; Little, 1995).

**Methodology:** The columns of the data matrix $X^*$ are assumed to represent ordered time points $t = 0, 1, ..., T-1$. For each subject (row) $i$, a random dropout time $t_{0,i}$ is sampled. All observations for that subject from time $t_{0,i}$ onwards are masked as missing.

$$M_{ij} = 0 \quad \forall j \geq t_{0,i}$$

**Implementation Details:** For each row $i$, the dropout time $t_{0,i}$ is sampled uniformly from the range of possible time steps, i.e., $t_{0,i} \sim \text{Unif}\{1, ..., T\}$.

### A.2.6 POLARIZATION-MNAR

**Description:** This pattern simulates data where values falling in the middle of a feature's distribution are preferentially removed, simulating survey non-response from individuals with moderate opinions. This is implemented by setting values between the $q$-th and $(1-q)$-th quantiles to missing. A "soft" version makes the observation probability proportional to the value's distance from the median. Polarization-MNAR is recognized in the survey methodology literature as a form of voluntary response bias or self-selection bias (Bethlehem, 2010). One obvious example can be found in online product reviews, which frequently exhibit a U-shaped or J-shaped distribution, heavily skewed towards the highest and lowest ratings while the middle ground remains sparse—sometimes referred to as the "brag-and-moan" bias (Hu et al., 2009). Similarly, in political polling, nonresponse bias can be exacerbated if highly polarized individuals are more motivated to participate than moderates, potentially exaggerating measures of mass polarization Cavari & Freedman (2023).

This pattern maps to the "Unstructured" MNAR categories of (Jackson et al., 2023). Specifically, our hard polarization variant, where values within a quantile range are missing with certainty, is an instance of **MNAR-UD** (Unstructured, Deterministic). Our soft polarization variant, where the missingness probability is a function of the value's distance from the median, is an example of **MNAR-UP** (Unstructured, Probabilistic).

**Hard Polarization Methodology:** For each column $j$, values falling between two quantiles are deterministically masked. The quantiles are calculated using only the set of currently observed (non-'NaN') values in that column. Let $X^*_{:,j}$ denote this set of observed values, i.e., $X^*_{:,j} = \{X_{ij} \mid M_{ij} = 1\}$. The lower and upper thresholds, $L_j$ and $H_j$, are defined as:

$$L_j = \text{quantile}(X^*_{:,j}, q_{\text{thresh}})$$

$$H_j = \text{quantile}(X^*_{:,j}, 1 - q_{\text{thresh}})$$

An entry $X_{ij}$ is then masked if its value falls between these two scalar thresholds:

$$M_{ij} = 0 \quad \text{if} \quad L_j < X_{ij} < H_j$$

**Soft Polarization Methodology:** The probability of a value being observed is made proportional to its normalized absolute distance from the column's median, $\mu_j$. This creates a softer, probabilistic version of the polarization effect. The missing probability is given by:

$$\mathbb{P}(M_{ij} = 0) = \epsilon + (1 - 2\epsilon) \frac{|X^*_{ij} - \mu_j|^\alpha}{\max_k(|X^*_{kj} - \mu_j|^\alpha)}$$

**Implementation Details:** For the hard polarization pattern, we introduce a hyperparameter $q_{\text{thresh}}$ for the threshold quantile that defines the central portion of the distribution to be masked. For the soft polarization pattern, we have an exponent parameter $\alpha$ that controls the intensity of the polarization effect. Higher values of $\alpha$ make the observation probability more sensitive to deviations from the median. In the soft version, we also have a baseline probability $\epsilon$ that ensures even values at the median have a non-zero chance of being observed.

### A.2.7 LATENT-FACTOR-MNAR

**Description:** This pattern generates a complex missingness structure where the probability of an entry being missing depends on unobserved (latent) characteristics of both its row and its column. This is most common in recommender systems and collaborative filtering, where the data (e.g., user ratings) are inherently MNAR (Marlin & Zemel, 2009; Steck, 2010). Users do not select items to rate uniformly; rather, their exposure to items and their decisions to provide a rating are strongly influenced by their underlying (latent) preferences-the very values the system aims to predict (Schnabel et al., 2016). This creates a selection bias where observed ratings are a biased sample of the full data. A standard approach in the literature assumes a low-rank structure for both the underlying data matrix and the missingness mechanism (propensity score matrix), implying they are governed by a shared set of latent factor (Sportisse et al., 2020a; Jin et al., 2022). Recent work in causal matrix completion explicitly models these latent confounders (Agarwal et al., 2023).

**Methodology:** The probability of an entry $(i, j)$ being observed is modeled using a low-rank bi-linear model. The observation probability is given by the sigmoid of a dot product of latent factors plus bias terms:

$$\mathbb{P}(M_{ij} = 1) = \sigma(u_i^T v_j + b_i + c_j)$$

where $u_i \in \mathbb{R}^k$ and $v_j \in \mathbb{R}^k$ are $k$-dimensional latent vectors for row $i$ and column $j$, and $b_i$ and $c_j$ are scalar biases for the row and column, respectively.

**Implementation Details:** We specify the rank $k$ that defines the dimensionality of the latent space, sampled as an integer. The elements of the latent factor matrices $U \in \mathbb{R}^{N \times k}$ and $V \in \mathbb{R}^{D \times k}$, and the bias vectors $b \in \mathbb{R}^N$ and $c \in \mathbb{R}^D$, are sampled independently from a standard normal distributions.

### A.2.8 CLUSTER-MNAR

**Description:** This pattern induces missingness based on latent group-level characteristics. Rows and columns are first assigned to discrete clusters, and each cluster has a random effect that uniformly influences the observation probability of all its members. Such structures are prevalent in education (students clustered in schools), healthcare (patients clustered in hospitals), and cross-national surveys. In these settings, missing data patterns are often not independent across observations but rather clustered (Van Buuren, 2011). For example, in multi-center clinical trials or Cluster Randomized Trials, missingness can be heavily influenced by site-specific factors such as resource availability or staff training (Diaz-Ordaz et al., 2014). Ignoring this hierarchical structure during imputation fails to account for the within-cluster correlation and can lead to biased estimates (Enders et al., 2016). Because the missingness mechanis is related to these cluster-level effects, which may themselves be latent, the data is MNAR.

In the structured missingness taxonomy, we classify Cluster-MNAR as **MNAR-UP** (Unstructured, Probabilistic) due to the the probabilistic dependence on unobserved factors, without dependence on other missingness indicators.

**Methodology:** The probability of an entry $(i, j)$ being observed is determined by an additive model of random effects corresponding to the cluster assignments of its row $i$ and column $j$. Denoting the row assignments by $C_R(i)$ and column assignments by $C_C(j)$, the observation probability is modeled as:

$$\mathbb{P}(M_{ij} = 1) = \sigma(g_{C_R(i)} + h_{C_C(j)} + \epsilon_{ij})$$

where:

- $\sigma(z) = (1 + e^{-z})^{-1}$ is the sigmoid function.

- $g_k \sim \mathcal{N}(0, \tau_r^2)$ is the random effect for row cluster $k$.
- $h_l \sim \mathcal{N}(0, \tau_c^2)$ is the random effect for column cluster $l$.
- $\epsilon_{ij} \sim \mathcal{N}(0, \epsilon_{\text{std}}^2)$ is an entry-specific noise term.

**Implementation Details:** For a matrix with $N$ and $D$ columns, row assignments $C_R(i)$ are draw uniformly from $\{0, ..., K_R - 1\}$ and column assignments $C_C(j)$ are drawn uniformly from $\{0, ..., K_C - 1\}$, where $K_R$ and $K_C$ are the total number of row and column clusters, respectively. The number of row clusters $K_R$, the number of column clusters $K_C$, and the standard deviation of the random effects ($\tau_r$, $\tau_c$, $\epsilon_{\text{std}}$) are hyperparameters specified for the data generation process.

### A.2.9   Two-Phase-MNAR

**Description:** This pattern mimics multi-stage data collection, mirroring established methodologies such as "two-phase sampling" (or "double sampling") (Neyman, 1938) and "Planned Missing Data Designs" (Graham et al., 2006). These designs are frequently employed in large-scale surveys and epidemiological studies to manage costs and participant burden when certain variables are expensive or difficult to measure (Rhemtulla & Hancock, 2016). Another example includes market research where basic demographics are collected from all participants, but detailed purchasing behavior is only gathered from a subset, with missingness related to income level. In these designs, a full sample provides baseline information ("cheap" features), and a subset is strategically selected for follow-up ("expensive" features).

**Methodology:** Let $\mathcal{F} = \{0, 1, ..., D - 1\}$ be the set of all column indices in the data matrix $X$. This set is randomly partitioned into a "cheap" subset $\mathcal{C} \subset \mathcal{F}$ and an "expensive" subset $\mathcal{E} \subset \mathcal{F}$, such that $\mathcal{C} \cup \mathcal{E} = \mathcal{F}$ and $\mathcal{C} \cap \mathcal{E} = \emptyset$. By design, features in the cheap set $\mathcal{C}$ are always observed.

The decision to collect the expensive features for a given row $i$ is based on a logistic model applied to its cheap features. Let $X_{i,\mathcal{C}}$ denote the vector of values $\{X_{ij} \mid j \in \mathcal{C}\}$ for row $i$. A score is calculated for each row:

$$s_i = \text{normalize}(X_{i,\mathcal{C}}^T w)$$

where $w$ is a vector of random weights and the 'normalize' function applies z-score normalization to the resulting scores across all rows.

The probability that all expensive features are observed for row $i$ is then given by:

$$\mathbb{P}(M_{ij} = 1 \text{ for all } j \in \mathcal{E}) = \sigma(\alpha + \beta \cdot s_i)$$

If the expensive features for row $i$ are not observed (based on the probability above), then all of its values in the expensive columns are masked as missing, i.e., $M_{ij} = 1$ for all $j \in \mathcal{E}$.

**Implementation Details:** A fraction of columns, e.g., 50%, are randomly assigned to be "cheap". The weight vector for the scoring model is sampled from a standard normal distribution, $w \sim \mathcal{N}(0, 1)$. Parameters $\alpha, \beta$ control the base rate and score-dependency of the observation probability. In our implementation, they are set to default values of $\alpha = 0$ and $\beta = 2.0$.

### A.3   Additional tables

Table 5: Other imputation methods

| Name | Description |
| --- | --- |
| Column-wise mean (Hawthorne & Elliott, 2005) | Mean of columns |
| SoftImpute (Hastie et al., 2015) | Iterative soft thresholding singular value decomposition based on a low-rank assumption on the data |
| $k$-Nearest Neighbors (Fix & Hodges, 1989) | Row-wise nearest neighbors mean |
| HyperImpute (Jarrett et al., 2022) | Iterative imputation method optimizing over a suite of imputation methods |
| Optimal transport method (Muzellec et al., 2020) | Uses optimal transport distances as a loss to impute missing values based on the principle that two randomly drawn batches from the same dataset should share similar data distributions |
| MissForest (Stekhoven & Bühlmann, 2011) | Repeatedly trains a random forest model for each variable on the observed values to predict and fill in missing entries until convergence |
| ICE (van Buuren & Groothuis-Oudshoorn, 2011) | Imputation with iterative and chained equations of linear/logistic models for conditional expectations |
| MICE (Royston & White, 2011) | Handles missing data by iteratively imputing each incomplete variable using regression models that condition on all other variables |
| GAIN (Yoon et al., 2018) | Adapts generative adversarial networks (Goodfellow et al., 2020) where the generator imputes missing values and the discriminator identifies which components are observed versus imputed |
| MIWAE (Mattei & Frellsen, 2019) | Learns a deep latent variable model and then performs importance sampling for imputation |
| ForestDiffusion (Jolicoeur-Martineau et al., 2024) | Trains a diffusion model using XGBoost directly on incomplete tabular data and then fills in missing values with an adapted inpainting algorithm |
| TabPFN | The `tabpfn-extensions` package includes a part to impute missing entries in a table column-by-column using TabPFN. |

Table 6: Imputation Accuracy ± Standard Error by Missingness Pattern (All Methods)

| Pattern | TabImpute | EWF-TabPFN | HyperImpute | OT | MissForest |
|---|---|---|---|---|---|
| MCAR | 0.882 ± 0.021 | 0.817 ± 0.024 | 0.804 ± 0.031 | 0.855 ± 0.019 | 0.867 ± 0.022 |
| NN-MNAR | 0.900 ± 0.019 | 0.876 ± 0.020 | 0.760 ± 0.036 | 0.814 ± 0.021 | 0.818 ± 0.029 |
| Self-Masking-MNAR | 0.703 ± 0.038 | 0.682 ± 0.042 | 0.689 ± 0.039 | 0.568 ± 0.037 | 0.638 ± 0.040 |
| Col-MAR | 0.877 ± 0.030 | 0.855 ± 0.027 | 0.814 ± 0.040 | 0.750 ± 0.040 | 0.773 ± 0.039 |
| Block-MNAR | 0.890 ± 0.026 | 0.905 ± 0.026 | 0.873 ± 0.027 | 0.857 ± 0.024 | 0.860 ± 0.023 |
| Seq-MNAR | 0.920 ± 0.014 | 0.905 ± 0.015 | 0.864 ± 0.030 | 0.894 ± 0.014 | 0.829 ± 0.031 |
| Panel-MNAR | 0.915 ± 0.025 | 0.711 ± 0.048 | 0.865 ± 0.033 | 0.887 ± 0.025 | 0.912 ± 0.020 |
| Polarization-MNAR | 0.804 ± 0.024 | 0.879 ± 0.021 | 0.622 ± 0.039 | 0.802 ± 0.020 | 0.567 ± 0.033 |
| Soft-Polarization-MNAR | 0.759 ± 0.045 | 0.864 ± 0.024 | 0.711 ± 0.034 | 0.750 ± 0.029 | 0.667 ± 0.041 |
| Latent-Factor-MNAR | 0.891 ± 0.021 | 0.883 ± 0.018 | 0.775 ± 0.036 | 0.842 ± 0.026 | 0.834 ± 0.023 |
| Cluster-MNAR | 0.903 ± 0.014 | 0.871 ± 0.021 | 0.839 ± 0.028 | 0.828 ± 0.022 | 0.830 ± 0.021 |
| Two-Phase-MNAR | 0.886 ± 0.026 | 0.851 ± 0.029 | 0.865 ± 0.031 | 0.807 ± 0.028 | 0.873 ± 0.020 |
| Censoring-MNAR | 0.710 ± 0.032 | 0.593 ± 0.045 | 0.786 ± 0.041 | 0.589 ± 0.036 | 0.671 ± 0.038 |
| Overall | 0.849 ± 0.008 | 0.823 ± 0.009 | 0.790 ± 0.010 | 0.788 ± 0.009 | 0.780 ± 0.009 |

| Pattern | K-Nearest Neighbors | ICE | SoftImpute | Col Mean | DiffPuter |
|---|---|---|---|---|---|
| MCAR | 0.750 ± 0.025 | 0.659 ± 0.041 | 0.617 ± 0.044 | 0.478 ± 0.041 | 0.478 ± 0.041 |
| NN-MNAR | 0.741 ± 0.026 | 0.627 ± 0.039 | 0.679 ± 0.035 | 0.497 ± 0.041 | 0.497 ± 0.041 |
| Self-Masking-MNAR | 0.614 ± 0.038 | 0.673 ± 0.049 | 0.396 ± 0.044 | 0.293 ± 0.038 | 0.293 ± 0.038 |
| Col-MAR | 0.839 ± 0.028 | 0.825 ± 0.038 | 0.703 ± 0.040 | 0.622 ± 0.046 | 0.598 ± 0.047 |
| Block-MNAR | 0.860 ± 0.028 | 0.812 ± 0.037 | 0.799 ± 0.042 | 0.782 ± 0.034 | 0.806 ± 0.029 |
| Seq-MNAR | 0.885 ± 0.018 | 0.815 ± 0.037 | 0.783 ± 0.036 | 0.775 ± 0.033 | 0.775 ± 0.033 |
| Panel-MNAR | 0.899 ± 0.028 | 0.797 ± 0.044 | 0.340 ± 0.053 | 0.787 ± 0.037 | 0.811 ± 0.032 |
| Polarization-MNAR | 0.641 ± 0.028 | 0.560 ± 0.041 | 0.665 ± 0.048 | 0.964 ± 0.012 | 0.964 ± 0.012 |
| Soft-Polarization-MNAR | 0.580 ± 0.042 | 0.729 ± 0.031 | 0.677 ± 0.033 | 0.747 ± 0.030 | 0.747 ± 0.030 |
| Latent-Factor-MNAR | 0.764 ± 0.029 | 0.703 ± 0.042 | 0.714 ± 0.039 | 0.590 ± 0.045 | 0.590 ± 0.045 |
| Cluster-MNAR | 0.774 ± 0.024 | 0.750 ± 0.042 | 0.627 ± 0.046 | 0.558 ± 0.044 | 0.558 ± 0.044 |
| Two-Phase-MNAR | 0.898 ± 0.020 | 0.876 ± 0.032 | 0.678 ± 0.047 | 0.671 ± 0.042 | 0.671 ± 0.042 |
| Censoring-MNAR | 0.515 ± 0.047 | 0.866 ± 0.035 | 0.455 ± 0.054 | 0.346 ± 0.040 | 0.307 ± 0.037 |
| Overall | 0.751 ± 0.010 | 0.746 ± 0.011 | 0.626 ± 0.013 | 0.624 ± 0.013 | 0.623 ± 0.013 |

| Pattern | TabPFN | MICE | MIWAE | GAIN |
|---|---|---|---|---|
| MCAR | 0.415 ± 0.041 | 0.329 ± 0.044 | 0.254 ± 0.043 | 0.545 ± 0.039 |
| NN-MNAR | 0.428 ± 0.041 | 0.300 ± 0.049 | 0.230 ± 0.043 | 0.392 ± 0.049 |
| Self-Masking-MNAR | 0.268 ± 0.037 | 0.611 ± 0.051 | 0.217 ± 0.041 | 0.452 ± 0.056 |
| Col-MAR | 0.549 ± 0.050 | 0.520 ± 0.054 | 0.488 ± 0.048 | 0.275 ± 0.054 |
| Block-MNAR | 0.582 ± 0.051 | 0.502 ± 0.048 | 0.606 ± 0.039 | 0.202 ± 0.051 |
| Seq-MNAR | 0.638 ± 0.045 | 0.475 ± 0.050 | 0.584 ± 0.045 | 0.204 ± 0.048 |
| Panel-MNAR | 0.234 ± 0.042 | 0.553 ± 0.048 | 0.269 ± 0.045 | 0.536 ± 0.059 |
| Polarization-MNAR | 0.907 ± 0.029 | 0.192 ± 0.037 | 0.562 ± 0.027 | 0.294 ± 0.045 |
| Soft-Polarization-MNAR | 0.600 ± 0.050 | 0.219 ± 0.038 | 0.548 ± 0.036 | 0.425 ± 0.062 |
| Latent-Factor-MNAR | 0.499 ± 0.048 | 0.338 ± 0.047 | 0.355 ± 0.045 | 0.296 ± 0.046 |
| Cluster-MNAR | 0.499 ± 0.047 | 0.327 ± 0.044 | 0.330 ± 0.044 | 0.357 ± 0.048 |
| Two-Phase-MNAR | 0.478 ± 0.049 | 0.564 ± 0.048 | 0.529 ± 0.049 | 0.228 ± 0.049 |
| Censoring-MNAR | 0.342 ± 0.041 | 0.648 ± 0.050 | 0.308 ± 0.045 | 0.320 ± 0.051 |
| Overall | 0.495 ± 0.014 | 0.429 ± 0.014 | 0.407 ± 0.013 | 0.348 ± 0.015 |

Table 7: Mean Column-wise $R^2$ ± Standard Error by Missingness Pattern

| Pattern | EWF-TabPFN | TabImpute | HyperImpute | ICE | MissForest |
|---|---|---|---|---|---|
| MCAR | 0.395 ± 0.036 | **0.414 ± 0.035** | 0.409 ± 0.035 | 0.387 ± 0.035 | 0.401 ± 0.036 |
| NN-MNAR | **0.387 ± 0.035** | 0.382 ± 0.034 | 0.364 ± 0.034 | 0.348 ± 0.033 | 0.351 ± 0.033 |
| Self-Masking-MNAR | **0.297 ± 0.039** | 0.285 ± 0.039 | 0.234 ± 0.038 | 0.250 ± 0.037 | 0.207 ± 0.030 |
| Col-MAR | **0.308 ± 0.033** | 0.289 ± 0.035 | 0.285 ± 0.034 | 0.287 ± 0.034 | 0.242 ± 0.032 |
| Block-MNAR | **0.283 ± 0.032** | 0.256 ± 0.033 | 0.242 ± 0.034 | 0.229 ± 0.033 | 0.221 ± 0.033 |
| Seq-MNAR | 0.234 ± 0.030 | 0.218 ± 0.027 | **0.246 ± 0.029** | 0.234 ± 0.030 | 0.210 ± 0.028 |
| Panel-MNAR | 0.275 ± 0.028 | **0.290 ± 0.029** | 0.285 ± 0.027 | 0.271 ± 0.029 | 0.271 ± 0.028 |
| Polarization-MNAR | **0.230 ± 0.022** | 0.190 ± 0.019 | 0.189 ± 0.020 | 0.177 ± 0.021 | 0.156 ± 0.017 |
| Soft-Polarization-MNAR | **0.177 ± 0.027** | 0.148 ± 0.024 | 0.146 ± 0.024 | 0.137 ± 0.023 | 0.131 ± 0.022 |
| Latent-Factor-MNAR | **0.347 ± 0.034** | 0.338 ± 0.033 | 0.335 ± 0.034 | 0.331 ± 0.033 | 0.322 ± 0.034 |
| Cluster-MNAR | **0.350 ± 0.036** | 0.342 ± 0.035 | 0.342 ± 0.037 | 0.325 ± 0.036 | 0.325 ± 0.036 |
| Two-Phase-MNAR | **0.370 ± 0.035** | 0.331 ± 0.037 | 0.333 ± 0.037 | 0.334 ± 0.036 | 0.310 ± 0.034 |
| Censoring-MNAR | **0.177 ± 0.025** | 0.143 ± 0.020 | 0.121 ± 0.019 | 0.125 ± 0.019 | 0.097 ± 0.017 |
| Overall | **0.295 ± 0.009** | 0.279 ± 0.009 | 0.272 ± 0.009 | 0.264 ± 0.009 | 0.249 ± 0.009 |

| Pattern | OT | K-Nearest Neighbors | MICE | GAIN | SoftImpute |
|---|---|---|---|---|---|
| MCAR | **0.401 ± 0.034** | 0.323 ± 0.036 | 0.302 ± 0.036 | 0.318 ± 0.031 | 0.265 ± 0.041 |
| NN-MNAR | **0.334 ± 0.033** | 0.278 ± 0.031 | 0.254 ± 0.033 | 0.265 ± 0.026 | 0.223 ± 0.043 |
| Self-Masking-MNAR | 0.211 ± 0.034 | 0.207 ± 0.033 | **0.264 ± 0.044** | 0.224 ± 0.029 | 0.166 ± 0.030 |
| Col-MAR | 0.221 ± 0.031 | **0.227 ± 0.032** | 0.182 ± 0.030 | 0.156 ± 0.027 | 0.167 ± 0.033 |
| Block-MNAR | **0.199 ± 0.029** | 0.186 ± 0.029 | 0.155 ± 0.027 | 0.098 ± 0.016 | 0.165 ± 0.035 |
| Seq-MNAR | 0.192 ± 0.023 | **0.213 ± 0.027** | 0.139 ± 0.023 | 0.099 ± 0.015 | 0.181 ± 0.032 |
| Panel-MNAR | 0.247 ± 0.025 | **0.277 ± 0.028** | 0.146 ± 0.021 | 0.167 ± 0.023 | 0.181 ± 0.032 |
| Polarization-MNAR | **0.177 ± 0.018** | 0.146 ± 0.018 | 0.109 ± 0.015 | 0.109 ± 0.013 | 0.118 ± 0.021 |
| Soft-Polarization-MNAR | **0.108 ± 0.020** | 0.075 ± 0.018 | 0.066 ± 0.016 | 0.078 ± 0.014 | 0.077 ± 0.021 |
| Latent-Factor-MNAR | **0.306 ± 0.031** | 0.246 ± 0.031 | 0.227 ± 0.032 | 0.202 ± 0.023 | 0.194 ± 0.039 |
| Cluster-MNAR | **0.314 ± 0.033** | 0.254 ± 0.032 | 0.225 ± 0.032 | 0.239 ± 0.029 | 0.187 ± 0.038 |
| Two-Phase-MNAR | 0.260 ± 0.030 | **0.293 ± 0.031** | 0.191 ± 0.030 | 0.198 ± 0.026 | 0.190 ± 0.035 |
| Censoring-MNAR | 0.084 ± 0.016 | 0.095 ± 0.011 | **0.131 ± 0.022** | 0.127 ± 0.020 | 0.081 ± 0.021 |
| Overall | **0.235 ± 0.009** | 0.217 ± 0.008 | 0.184 ± 0.008 | 0.175 ± 0.007 | 0.169 ± 0.009 |

| Pattern | ForestDiffusion | TabPFN | MIWAE | Col Mean |
|---|---|---|---|---|
| MCAR | **0.220 ± 0.029** | 0.070 ± 0.008 | 0.036 ± 0.005 | 0.000 ± 0.000 |
| NN-MNAR | **0.177 ± 0.026** | 0.042 ± 0.006 | 0.025 ± 0.003 | 0.000 ± 0.000 |
| Self-Masking-MNAR | **0.116 ± 0.024** | 0.106 ± 0.016 | 0.026 ± 0.004 | 0.000 ± 0.000 |
| Col-MAR | **0.108 ± 0.020** | 0.081 ± 0.011 | 0.030 ± 0.003 | 0.000 ± 0.000 |
| Block-MNAR | **0.106 ± 0.017** | 0.051 ± 0.008 | 0.023 ± 0.003 | 0.000 ± 0.000 |
| Seq-MNAR | **0.089 ± 0.014** | 0.034 ± 0.003 | 0.022 ± 0.003 | 0.000 ± 0.000 |
| Panel-MNAR | **0.120 ± 0.016** | 0.083 ± 0.011 | 0.039 ± 0.006 | 0.000 ± 0.000 |
| Polarization-MNAR | **0.091 ± 0.012** | 0.036 ± 0.004 | 0.029 ± 0.004 | 0.000 ± 0.000 |
| Soft-Polarization-MNAR | **0.031 ± 0.004** | 0.004 ± 0.001 | 0.014 ± 0.001 | 0.000 ± 0.000 |
| Latent-Factor-MNAR | **0.152 ± 0.022** | 0.053 ± 0.007 | 0.028 ± 0.003 | 0.000 ± 0.000 |
| Cluster-MNAR | **0.181 ± 0.027** | 0.058 ± 0.009 | 0.033 ± 0.005 | 0.000 ± 0.000 |
| Two-Phase-MNAR | **0.126 ± 0.022** | 0.086 ± 0.012 | 0.022 ± 0.003 | 0.000 ± 0.000 |
| Censoring-MNAR | 0.071 ± 0.011 | **0.072 ± 0.018** | 0.067 ± 0.018 | 0.000 ± 0.000 |
| Overall | **0.122 ± 0.006** | 0.060 ± 0.003 | 0.030 ± 0.002 | 0.000 ± 0.000 |

Table 8: Mean Normalized Negative RMSE ± Standard Error by Missingness Pattern. Note that these normalized values are slightly different than Tab. 1 because the nonlinear model is included here.

| Pattern | TabImpute | HyperImpute | TabImpute$_{\text{Nonlinear}}$ | MissForest |
|---|---|---|---|---|
| MCAR | **0.880 ± 0.021** | 0.803 ± 0.031 | 0.835 ± 0.022 | 0.866 ± 0.022 |
| NN-MNAR | **0.898 ± 0.019** | 0.758 ± 0.036 | 0.835 ± 0.022 | 0.816 ± 0.029 |
| Self-Masking-MNAR | **0.705 ± 0.037** | 0.689 ± 0.039 | 0.613 ± 0.043 | 0.639 ± 0.041 |
| Col-MAR | **0.868 ± 0.030** | 0.806 ± 0.040 | 0.836 ± 0.029 | 0.764 ± 0.038 |
| Block-MNAR | **0.890 ± 0.026** | 0.873 ± 0.027 | 0.868 ± 0.029 | 0.860 ± 0.023 |
| Seq-MNAR | **0.919 ± 0.015** | 0.863 ± 0.030 | 0.890 ± 0.018 | 0.828 ± 0.031 |
| Panel-MNAR | **0.893 ± 0.027** | 0.847 ± 0.034 | 0.448 ± 0.063 | 0.891 ± 0.022 |
| Polarization-MNAR | **0.800 ± 0.024** | 0.619 ± 0.039 | 0.751 ± 0.030 | 0.564 ± 0.032 |
| Soft-Polarization-MNAR | 0.756 ± 0.045 | 0.709 ± 0.034 | **0.808 ± 0.034** | 0.665 ± 0.041 |
| Latent-Factor-MNAR | **0.888 ± 0.020** | 0.773 ± 0.035 | 0.857 ± 0.020 | 0.832 ± 0.024 |
| Cluster-MNAR | **0.896 ± 0.016** | 0.833 ± 0.029 | 0.858 ± 0.017 | 0.824 ± 0.022 |
| Two-Phase-MNAR | **0.879 ± 0.026** | 0.859 ± 0.031 | 0.871 ± 0.026 | 0.867 ± 0.020 |
| Censoring-MNAR | 0.711 ± 0.032 | **0.790 ± 0.039** | 0.623 ± 0.039 | 0.673 ± 0.036 |
| Overall | **0.845 ± 0.008** | 0.786 ± 0.010 | 0.776 ± 0.010 | 0.776 ± 0.009 |

Table 9: Synthetic Data Generation Parameters

| Missingness Pattern | Parameter Name | Symbol | Value |
|---|---|---|---|
| MCAR | Missing probability | $p$ | 0.4 |
| Col-MAR | Missing probability | $p$ | 0.4 |
| NN-MNAR | Neighborhood size | $|N_{ij}|$ | Variable |
| | Network layers | $L$ | Random |
| | Network depth | $d$ | Random |
| | Weight initialization | $W$ | Random |
| | Bias initialization | $b$ | Random |
| Self-Masking-MNAR | Coefficient set | $\alpha$ | $\{-2, -1, 1, 2\}$ |
| | Target missing proportion | $p_{missing}$ | Variable |
| Block-MNAR | Missing probability | $p$ | 0.4 |
| | Matrix size N | $N$ | 100 |
| | Matrix size T | $T$ | 50 |
| | Row blocks | $B_r$ | 10 |
| | Column blocks | $B_c$ | 10 |
| | Convolution type | - | mean |
| Seq-MNAR | Missing probability | $p$ | 0.4 |
| | Algorithm | - | epsilon_greedy |
| | Pooling | - | False |
| | Epsilon | $\epsilon$ | 0.4 |
| | Epsilon decay | $\gamma$ | 0.99 |
| | Random seed | $s$ | 42 |
| Panel-MNAR | No explicit hyperparameters (dropout time sampled uniformly) | | |
| Polarization-MNAR | Threshold quantile | $q_{thresh}$ | 0.25 |
| Soft-Polarization-MNAR | Polarization alpha | $\alpha$ | 2.5 |
| | Polarization epsilon | $\epsilon$ | 0.05 |
| Latent-Factor-MNAR | Latent rank (low) | $k_{low}$ | 1 |
| | Latent rank (high) | $k_{high}$ | 5 |
| Cluster-MNAR | Number of row clusters | $K_R$ | 5 |
| | Number of column clusters | $K_C$ | 4 |
| Two-Phase-MNAR | Cheap feature fraction | $f_{cheap}$ | 0.4 |
| Censoring-MNAR | Censoring quantile | $q_{censor}$ | 0.25 |

Table 10: OpenML datasets

| Dataset | Size | Domain | Description |
|---|---|---|---|
| EgyptianSkulls | $150 \times 5$ | Anthropology | Cranial measurements over time in Egypt |
| humans_numeric | $75 \times 15$ | Biology | Human body measurements |
| FacultySalaries | $50 \times 5$ | Education/Economics | University faculty salary data |
| SMSA | $59 \times 16$ | Demographics/Economics | U.S. metropolitan statistical area data |
| Student-Scores | $56 \times 13$ | Education | Student exam scores |
| analcatdata_election2000 | $67 \times 15$ | Political science | 2000 U.S. presidential election results |
| analcatdata_gviolence | $74 \times 9$ | Criminology | Gun violence statistics |
| analcatdata_olympic2000 | $66 \times 12$ | Sports/Economics | Olympic results and country stats |
| baskball | $96 \times 5$ | Sports analytics | Basketball performance data |
| visualizing_hamster | $73 \times 6$ | Education/Toy | Example dataset for teaching |
| witmer_census_1980 | $50 \times 5$ | Demographics | U.S. census microdata (1980) |
| MercuryinBass | $53 \times 10$ | Environmental chemistry | Mercury concentrations in fish |
| SolarPower | $204 \times 5$ | Energy/Engineering | Solar power output records |
| WineDataset | $178 \times 14$ | Chemistry/Oenology | Wine physicochemical properties |
| alcohol-qcm-sensor | $125 \times 15$ | Analytical chemistry | Alcohol detection sensor readings |
| benzo32 | $195 \times 33$ | Chemistry/Toxicology | Benzodiazepine compound data |
| machine_cpu | $209 \times 7$ | Computer systems | Predicting CPU performance |
| pwLinear | $200 \times 11$ | Mathematics/Engineering | Piecewise linear regression benchmark |
| pyrim | $74 \times 28$ | Chemistry/Pharmacology | Pyrimethamine bioassay compounds |
| slump | $103 \times 10$ | Civil engineering | Concrete slump test properties |
| ICU | $200 \times 20$ | Medicine | Intensive care patient data |
| appendicitis_test | $106 \times 8$ | Medicine | Appendicitis diagnosis |
| appendicitis_test_edsa | $106 \times 8$ | Medicine | Educational appendicitis dataset |
| breast-cancer-coimbra | $116 \times 10$ | Medicine | Breast cancer diagnosis data |
| Rainfall-in-Kerala-1901-2017 | $117 \times 18$ | Climate science | Rainfall time series in Kerala |
| pollution | $60 \times 16$ | Environmental science | Air pollution measurements |
| treepipit | $86 \times 10$ | Ecology | Bird habitat distribution |
| autoPrice | $159 \times 16$ | Business/Economics | Automobile pricing dataset |
| dataset_analcatdata_creditscore | $100 \times 7$ | Finance | Credit scoring dataset |
| Swiss-banknote-conterfeit-detection | $200 \times 7$ | Finance/Fraud | Banknote authenticity classification |
| Glass-Classification | $214 \times 10$ | Forensics/Materials | Glass chemical composition (forensics) |
| chatfield_4 | $235 \times 13$ | Statistics/Time series | Textbook time series data (Chatfield) |
| chscase_vine1 | $52 \times 10$ | Agriculture/Statistics | Vine growth study |
| edm | $154 \times 18$ | Education | Student learning performance |
| metafeatures | $75 \times 32$ | Meta-learning | Dataset-level features |
| rabe_131 | $50 \times 6$ | Chemistry/Benchmark | Spectroscopy regression dataset |
| rabe_148 | $66 \times 6$ | Chemistry/Benchmark | Spectroscopy regression dataset |
| rabe_265 | $51 \times 7$ | Chemistry/Benchmark | Spectroscopy regression dataset |
| sleuth_case1201 | $50 \times 7$ | Statistics/Education | Applied regression textbook data |
| sleuth_ex1605 | $62 \times 6$ | Statistics/Education | Applied regression textbook data |
| wisconsin | $194 \times 33$ | Medicine | Wisconsin breast cancer dataset |

Table 11: Non-normalized RMSE values for MCAR pattern by dataset. Dataset columns are standardized based on observed values (mean 0, variance 1).

| Dataset | TabImpute | EWF-TabPFN | HyperImpute | MissForest |
|---|---|---|---|---|
| EgyptianSkulls | **0.901** | 0.917 | 0.961 | 0.946 |
| FacultySalaries | 0.607 | 0.693 | 0.811 | **0.567** |
| Glass-Classification | 0.889 | 0.937 | **0.810** | 0.886 |
| ICU | 1.059 | **1.045** | 1.137 | 1.112 |
| MercuryinBass | **1.311** | 1.373 | 1.377 | 1.343 |
| Rainfall-in-Kerala-1901-2017 | 0.908 | 0.944 | **0.902** | 0.950 |
| SMSA | 0.840 | 0.814 | 0.939 | **0.786** |
| SolarPower | 0.889 | **0.805** | 0.873 | 0.888 |
| Student-Scores | 0.429 | **0.392** | 0.441 | 0.435 |
| Swiss-banknote-conterfeit-detection | 0.774 | 0.976 | 0.856 | **0.730** |
| WineDataset | 0.829 | 0.928 | 0.803 | **0.797** |
| alcohol-qcm-sensor | **0.467** | 0.590 | 0.580 | 0.517 |
| analcatdata_election2000 | **0.364** | 0.538 | 0.662 | 0.615 |
| analcatdata_gviolence | **0.687** | 0.796 | 0.801 | 0.877 |
| analcatdata_olympic2000 | 1.950 | **1.871** | 1.922 | 1.895 |
| appendicitis_test | 0.694 | 0.861 | **0.650** | 0.792 |
| appendicitis_test_edsa | 0.534 | 0.668 | **0.532** | 0.569 |
| autoPrice | 0.735 | 0.834 | 0.762 | **0.702** |
| baskball | **1.008** | 1.028 | 1.100 | 1.109 |
| benzo32 | 1.070 | 1.055 | 1.055 | **1.053** |
| breast-cancer-coimbra | 1.068 | **1.058** | 1.183 | 1.098 |
| chatfield_4 | **0.473** | 0.476 | 0.531 | 0.492 |
| chscase_vine1 | 0.928 | **0.809** | 0.996 | 0.833 |
| dataset_analcatdata_creditscore | 1.238 | **1.063** | 1.131 | 1.122 |
| divorce_prediction | 0.542 | 0.525 | 0.528 | **0.467** |
| edm | 0.612 | 0.763 | **0.553** | 0.566 |
| humans_numeric | 0.989 | **0.952** | 0.970 | 0.991 |
| machine_cpu | 0.902 | 1.039 | 0.956 | **0.858** |
| metafeatures | 2.609 | 2.263 | **2.090** | 2.225 |
| pollution | 1.946 | 1.967 | 2.065 | **1.837** |
| pwLinear | **1.031** | 1.075 | 1.356 | 1.356 |
| pyrim | **0.847** | 0.877 | 0.863 | 0.859 |
| rabe_131 | 1.016 | **0.792** | 1.003 | 1.031 |
| rabe_148 | **0.905** | 0.985 | 1.017 | 0.958 |
| rabe_265 | 1.104 | 1.131 | 1.298 | **1.090** |
| sleuth_case1201 | 0.940 | 1.054 | **0.939** | 0.963 |
| sleuth_ex1605 | **1.026** | 1.087 | 1.257 | 1.071 |
| slump | 0.750 | 0.871 | **0.650** | 0.805 |
| treepipit | 1.226 | 1.265 | **1.169** | 1.172 |
| visualizing_hamster | 0.754 | 0.872 | 0.905 | **0.750** |
| wisconsin | 0.854 | 0.777 | **0.685** | 0.783 |
| witmer_census_1980 | 0.654 | 0.704 | **0.616** | 0.637 |

Table 12: Non-normalized RMSE Values for MCAR Pattern by Dataset including ReMasker. Note that we could not replicate ReMasker's results using their open-source implementation. In order to not misrepresent their method, we do not include this table in our main results.

| Dataset | TabImpute | HyperImpute | MissForest | ReMasker |
|---|---|---|---|---|
| EgyptianSkulls | **0.901** | 0.961 | 0.946 | 2.329 |
| FacultySalaries | 0.607 | 0.811 | **0.567** | 1.534 |
| Glass-Classification | 0.889 | **0.810** | 0.886 | 3.126 |
| ICU | **1.059** | 1.137 | 1.112 | 1.479 |
| MercuryinBass | **1.311** | 1.377 | 1.343 | 2.198 |
| Rainfall-in-Kerala-1901-2017 | 0.908 | **0.902** | 0.950 | 3.560 |
| SMSA | 0.840 | 0.939 | **0.786** | 1.979 |
| SolarPower | 0.889 | **0.873** | 0.888 | 1.589 |
| Student-Scores | **0.429** | 0.441 | 0.435 | 1.785 |
| Swiss-banknote-conterfeit-detection | 0.774 | 0.856 | **0.730** | 1.364 |
| WineDataset | 0.829 | 0.803 | **0.797** | 1.454 |
| alcohol-qcm-sensor | **0.467** | 0.580 | 0.517 | 2.044 |
| analcatdata_election2000 | **0.364** | 0.662 | 0.615 | 4.563 |
| analcatdata_gviolence | **0.687** | 0.801 | 0.877 | 1.617 |
| analcatdata_olympic2000 | 1.950 | 1.922 | **1.895** | 2.787 |
| appendicitis_test | 0.694 | **0.650** | 0.792 | 2.174 |
| appendicitis_test_edsa | 0.534 | **0.532** | 0.569 | 1.986 |
| autoPrice | 0.735 | 0.762 | **0.702** | 2.190 |
| baskball | **1.008** | 1.100 | 1.109 | 2.246 |
| benzo32 | 1.070 | 1.055 | **1.053** | 3.282 |
| breast-cancer-coimbra | **1.068** | 1.183 | 1.098 | 1.820 |
| chatfield_4 | **0.473** | 0.531 | 0.492 | 3.545 |
| chscase_vine1 | 0.928 | 0.996 | **0.833** | 2.720 |
| dataset_analcatdata_creditscore | 1.238 | 1.131 | **1.122** | 1.469 |
| divorce_prediction | 0.542 | 0.528 | **0.467** | 1.216 |
| edm | 0.612 | **0.553** | 0.566 | 1.415 |
| humans_numeric | 0.989 | **0.970** | 0.991 | 4.165 |
| machine_cpu | 0.902 | 0.956 | **0.858** | 1.402 |
| metafeatures | 2.609 | **2.090** | 2.225 | 2.715 |
| pollution | 1.946 | 2.065 | **1.837** | 2.985 |
| pwLinear | **1.031** | 1.356 | 1.356 | 1.040 |
| pyrim | **0.847** | 0.863 | 0.859 | 1.647 |
| rabe_131 | 1.016 | **1.003** | 1.031 | 2.076 |
| rabe_148 | **0.905** | 1.017 | 0.958 | 2.508 |
| rabe_265 | 1.104 | 1.298 | **1.090** | 2.633 |
| sleuth_case1201 | 0.940 | **0.939** | 0.963 | 2.455 |
| sleuth_ex1605 | **1.026** | 1.257 | 1.071 | 2.327 |
| slump | 0.750 | **0.650** | 0.805 | 1.960 |
| treepipit | 1.226 | **1.169** | 1.172 | 1.648 |
| visualizing_hamster | 0.754 | 0.905 | **0.750** | 1.884 |
| wisconsin | 0.854 | **0.685** | 0.783 | 2.422 |
| witmer_census_1980 | 0.654 | **0.616** | 0.637 | 2.487 |

