# OpenReview forum: "TabImpute: Accurate and Fast Zero-Shot Missing-Data Imputation with a Pre-Trained Transformer"
_ICLR.cc/2026/Conference — Submitted to ICLR 2026_

### Official Review · Reviewer_V9f3 · 2025-10-28

**Soundness:** 3
**Presentation:** 3
**Contribution:** 3
**Rating:** 6
**Confidence:** 4

**Summary:**

The paper proposes TabImpute, a pre-trained transformer model for zero-shot missing data imputation for tabular datasets.
TabImpute is based on another method called TabPFN (the first version is published 2022 and the final version with extended experiments is published in nature, 2025).
TabImpute leverages an entry-wise featurization that enables efficient and parallel imputation, overcoming major speed and scalability barriers of previous transformer-based approaches. The model is trained entirely on synthetically generated tabular data with realistic missingness patterns, including MCAR, MAR, and eleven detailed MNAR scenarios. The pipeline enables strong generalization.

The paper introduces MissBench, a benchmark suite consisting of 42 publicly available OpenML datasets and 13 missingness mechanisms spanning domains such as medicine, finance, and engineering. TabImpute is shown to outperform 11 well-established imputation methods and recent foundation models in terms of accuracy and runtime, especially for high missingness rates. The final method adaptively ensembles TabImpute with EWF-TabPFN to leverage complementary strengths, further improving accuracy.​

The entire framework, including model, training pipeline, and benchmark suite, is open-sourced. TabImpute addresses the key limitations of prior methods (slow tuning, narrow applicability) and provides a fast, accurate, and general-purpose solution for missing-value problems in modern machine learning workflows.​

The main novelties include 1) entry-wise featurization for tabular imputation, enabling parallel and scalable predictions with transformers.
2) Synthetic data generation pipeline supporting a wide variety of missingness patterns, especially MNAR. 3) Introduction of MissBench, a comprehensive benchmark for tabular imputation. 4) Adaptive ensemble (TabImpute + EWF-TabPFN) achieving state-of-the-art zero-shot imputation accuracy across datasets and missingness scenarios. 5) Open-source release of code, models, evaluation pipeline, and benchmark.

**Strengths:**

1. The method works fast and does not require any extra fitting or hyperparameter tuning, making it efficient for real-world applications.​​
2. It handles many types of missing data and still gives accurate results across different domains (results include evaluations on tabular datasets of medicine, finance, and engineering).​​
3. TabImpute was evaluated on a large benchmark (42 real datasets with 13 missingness patterns) and performed better than 11 other known imputation methods.​​
4. The authors shares their code and models openly, encouraging reproducibility and making it easy for others to build on or apply their work.

**Weaknesses:**

1. It seems that the scalability for large tables is limited. TabImpute’s transformer architecture, combined with the new entry-wise featurization, results in a quadratic time complexity with respect to both rows and columns, effectively squaring the computational cost compared to standard attention.
While the method works well for the relatively small tables in the benchmark (up to hundreds of rows and columns), it will likely struggle with larger, industry-scale datasets (thousands to millions of entries), both in terms of speed and memory.​

2. All experiments and methods focus strictly on numerical tables, ignoring the reality that most tabular datasets in medicine, business, and public policy contain categorical features or a mix of types. Compared to several strong baselines (e.g., MissForest, HyperImpute), which do support mixed types, TabImpute currently cannot be used “as is” on real-world data without preprocessing or architectural changes.​

3. The pre-training of TabImpute uses synthetic data generation based mainly on linear factor models, though it simulates various missingness types. Generalization to datasets whose structure (e.g., highly non-linear interactions or domain-specific distributions) differs from training priors may not be robust. The method’s performance on real, extremely complex medical or financial MNAR cases is not thoroughly validated, so claims of universal generalization remain partly unproven.​

4. The ensemble step that adaptively weights TabImpute and EWF-TabPFN based on error on observed entries, while clever for accuracy, could potentially bias imputations if observed values are not representative, especially in systematic MNAR settings not covered in training.​

5. The paper does not address failure modes or vulnerable cases. For instance the authros could address highly sparse matrices, blocks of missingness, or adversarially masked data where generative pre-trained models may break down.​

6. While TabImpute is described as “fast” and “efficient,” there is no comprehensive analysis of wall-clock time, GPU/CPU requirements in varied environments, or deployment cost compared to industry standards, apart from per-entry runtime benchmarks. For practical adoption, these details are essential.​

7. Removing the attention mask and recasting imputation as entry-wise prediction introduces new statistical dependencies. The paper claims “no data leakage,” but fails to fully analyze potential risks. For example the authors could  analyze using test data to inform train predictions.​

**Questions:**

1. Considering the quadratic complexity of both attention and entry-wise featurization, can the proposed TabImpute realistically scale to industry datasets with millions of rows or columns? What architectural or algorithmic innovations would be necessary to prevent memory and computation bottlenecks, and can you demonstrate such scaling beyond what is shown in MissBench?

2. How does TabImpute perform on highly non-linear, domain-specific real-world datasets whose data distributions, interactions, or missingness patterns diverge significantly from your synthetic training priors? Can you provide evidence or theory supporting robustness to out-of-distribution missingness in places like healthcare or finance, especially when the linear factor model and MNAR pattern simulations clearly do not match real-world generative processes?

3. TabImpute does not currently support categorical or mixed-type tabular data. What specific obstacles prevent extending your model to these widely-used formats, and how would your training and inference procedures need to change? Can you show that transformer-based imputation can compete with specialized classical methods (such as MissForest or HyperImpute) on mixed-type datasets without extensive feature engineering?

---

> ### Author Response · Authors · 2025-11-19
>
> Thank you for your careful and comprehensive review. We greatly appreciate your assessment of our method’s performance and MissBench. We address your comments and questions below and in our top-level comment:
>
> 1. We agree that the computational complexity is indeed a bottleneck of the architecture that inhibits our model from scaling to tables with thousands of entries. We detail exactly which component of our model leads to this and relevant future directions to fix it in the top-level comment. For now, imputation on larger tables would require doing so by submatrix.
>
> 2. Verifying non-linearity of real world datasets is a non-trivial task. Most of the literature we found deals with determining the rank of factor models or testing the existence of correlation across the residuals. If the reviewer could suggest a specific testing procedure we are not aware of, we are open to apply it to the OpenML dataset in our MissBench and provide the testing results.  Our newly trained model, using only the MCAR pattern for training, outperforms all the imputation techniques in our benchmark 8 times out of the 13 patterns (refer to the table). We expect MCAR to be too simple to explain the missing patterns of real-world dataset; despite its simplicity, we observe that feeding diverse MCAR patterns into the Transformer architecture does achieve significant extrapolation.
>
> 3. We agree that native support for categorical data is a crucial and natural next step. Although our current version of TabImpute can operate with categorical variables in certain ways (e.g., one-hot encoding during featurization), it is not natively designed to handle categorical variables. We do not anticipate bottlenecks for adapting TabImpute to categorical features and outputs (i.e. there are no fundamental obstacles preventing this adaptation), but there are several tasks that would first need to be completed beforehand. First, we would need to design a new synthetic data generation pipeline that includes categorical variables (along with one-hot encoding featurizations). Second, we would need to design/train two separate models (both capable of dealing with one-hot encoded features) with different heads allowing either numeric or categorical outputs.
>
> We hope we have addressed your questions. Thank you again for your valuable feedback and we look forward to further discussion.

---

> > ### Author Response · Authors · 2025-11-23
> >
> > Hello, and thanks again for the thoughtful reviews. We’ve uploaded our rebuttal and hope it resolves the concerns you raised. Please feel free to let us know if any additional clarification would be helpful.

---

> > > ### Comment · Reviewer_V9f3 · 2025-11-26
> > > **promising, but important limitations**
> > >
> > > Thank you for the detailed response. Your clarifications on scalability limits, the extrapolation from MCAR training to more complex patterns, and the planned extension to categorical data are helpful and make the scope and future directions of TabImpute much clearer. I think I will keep my score.

---

### Official Review · Reviewer_HSxa · 2025-10-30

**Soundness:** 2
**Presentation:** 2
**Contribution:** 2
**Rating:** 4
**Confidence:** 4

**Summary:**

This paper introduces TabImpute, a pre-trained transformer model for zero-shot missing data imputation in tabular datasets, by extending TabPFN with entry-wise featurization (EWF). This enables parallel GPU computation of missing values, achieving major speedup over TabPFN's column-wise approach. The model is trained on 25 million synthetic datasets generated using linear factor models with diverse missingness patterns. The authors also introduce MissBench, a benchmark with 42 OpenML datasets and 13 missingness patterns. Finally, they ensemble TabImpute and a pre-trained TabPFN to introduce TabImpute+.

**Strengths:**

1. The entry-wise featurization cleverly recasts PFNs for cell-level imputation, enabling efficient parallel computation.

2. The softmax-based reweighing scheme that dynamically adjusts missingness pattern proportions during training is principled and seems effective. Additionally, the closed-form solution for the adaptive ensembling is theoretically motivated and quite interesting.

3. The clearest contribution of this work is the MissBench dataset. The diversity of 13 MNAR patterns is particularly valuable given their prevalence in real-world applications. The limited set of standard evaluation conditions is a key limitation in current imputation literature.

**Weaknesses:**

1. The 13 MNAR patterns, while extensive, lack theoretical grounding. Notably, sequential missingness patterns (Seq-MNAR) are inappropriate for tabular data since columns are permutation-invariant. Such patterns only make sense for repeated measures of the same feature over time, which doesn't appear to be the case here. More critically, the paper fails to reference prior work on "structured missingness" [1,2], which defines missingness mechanisms orthogonal and complementary to the Rubin (1976) framework (MCAR/MAR/MNAR). Many of the paper's patterns (Block-MNAR, Panel-MNAR, Cluster-MNAR) align with existing structured missingness taxonomies but are presented without proper attribution. The authors should either: (a) adopt established structured missingness frameworks and clearly map their patterns to these taxonomies, or (b) provide stronger justification for why their patterns represent realistic real-world scenarios beyond the brief qualitative descriptions provided. This would strengthen the paper's foundation and clarify its relationship to prior work.

2. The claim that TabImpute+ achieves state-of-the-art performance is inadequately supported due to missing comparisons with recent imputation methods [3,4,5]. This omission is particularly concerning because [3] and [5] are transformer-based approaches that would provide fairer architectural comparisons. Currently, all baselines are non-transformer methods (random forests, GANs, matrix factorization, etc.). The authors should include these baselines or explicitly justify their exclusion to substantiate their state-of-the-art claims.

3. The results in Table 2 raise concerns about TabImpute+'s practical utility. For most individual patterns, either TabImpute or EWF-TabPFN alone achieves top performance. The "Overall" metric is misleadingly close since it's normalized over only these 3 methods, obscuring the fact that the ensemble provides minimal added value (also not statistically significant). More critically, TabImpute exhibits catastrophic failures on several patterns (Censoring-MNAR, Self-Masking-MNAR), while EWF-TabPFN maintains consistently reasonable performance across all patterns. For practitioners prioritizing robustness over marginal average gains, EWF-TabPFN appears to be the safer choice.

4. The O(n^2m + nm^2) complexity due to entry-wise featurization is prohibitive. The authors acknowledge this but provide no concrete solutions beyond "imputing in chunks". This fundamentally limits its utility to real-world imputation challenges (like biobanks).


#### References
[1] A Complete Characterization of Structured Missingness (2023)

[2] Learning from data with structured missingness (Nat. Mech. Int., 2023)

[3] CACTI: Leveraging Copy Masking and Contextual Information to Improve Tabular Data Imputation (ICML 2025)

[4] DiffPuter: Empowering Diffusion Models for Missing Data Imputation (ICLR 2025)

[5] ReMasker: Imputing Tabular Data with Masked Autoencoding (ICLR 2024)

**Questions:**

1. SDs are provided, but no significance testing is performed. Are TabImpute+'s improvements over HyperImpute/MissForest statistically significant? Also how were the SDs calculated for the normalized score? The cross method re-normizaiation is effected by the variance of the performance of each method.

2. What inductive biases allow a model trained on LFMs to work on non-linear real-world data? Is the transformer learning general imputation strategies, or is most of the performance on  non-linear settings originating from EWF-TabPFN?

3. Can the authors explain why TabImpute+ underperforms on Censoring-MNAR despite training including conceptually similar patterns?

4. Authors state they tried “nonlinear factor models and structural causal models (SCM) similar to the ones used in TabPFN”. However Table 5 does not include results for this? Am I missing something? could the authors make it clear which TabImpute models used SCMs?

---

> ### Author Response · Authors · 2025-11-19
>
> Thank you for your thorough and constructive feedback. We have addressed your comments below and incorporated your suggestions into our revised submission.
>
> Weaknesses:
> 1. We appreciate you directing us to the structured missingness (SM) literature. Our introduction has been updated to include a discussion of SM and its relationship to Rubin’s existing framework. We have expanded the Appendix with additional context and prior work motivating each of our missingness patterns. For example, we now reference the “brag-and-moan” bias in online product reviews [1] to contextualize the Polarization-MNAR pattern. Similarly, we now cite references to left-censored MNAR data in the proteomics literature [2] as motivation for the Censoring-MNAR pattern.  Where applicable, we have also mapped our patterns to the SM taxonomy. We note that some of our patterns, such as Latent-Factor-MNAR (where missingness depends on unobserved latent variables rather than just the values of X), fall outside the definition of SM provided in [3], and thus do not have a direct SM characterization.
>
> 2. We have included Diffputer in our updated benchmark results. For CACTI, we have omitted it from the benchmark primarily because it differs in problem scope: it relies on auxiliary metadata (column names and text descriptions) to leverage semantic context, whereas TabImpute and our baselines operate strictly on the observed data values. In addition, unlike our pre-trained zero-shot approach, CACTI requires dataset-specific training; retraining it across the 42 datasets and 13 missingness patterns unfortunately proved infeasible within the given timeframe. With respect to ReMasker, we evaluated it using the official GitHub implementation but were unable to replicate the reported performance. We placed these results in Appendix section A.3 in Table 11 instead of the main evaluation figure and tables to avoid misrepresenting the method's capabilities due to reproducibility issues.
>
> 3. We provide a more detailed analysis of this concern in the main rebuttal response. In summary, after fixing a critical normalization bug in our codebase, we observe significant improvement on MAR and MNAR patterns, including Censoring-MNAR and Self-Masking-MNAR. On patterns where EWF-TabPFN outperforms TabImpute,  the performance gap has narrowed considerably.
>
> 4. We acknowledge the computational complexity as a limitation of the current architecture. Addressing scalable architectures for tabular foundation models is an open and active area of research. An expanded discussion of this limitation and potential mitigation strategies (beyond chunking) can be found above in our main comment to reviewers.
>
> [1] Hu, Nan, Jie Zhang, and Paul A. Pavlou. "Overcoming the J-shaped distribution of product reviews."
>
> [2] Jackson, James, et al. "A complete characterisation of structured missingness."
>
> [3] Karpievitch, Yuliya, et al. "A statistical framework for protein quantitation in bottom-up MS-based proteomics."
>
> Questions:
> 1. We calculate standard deviations after normalization. We performed a paired t-test and a Wilcoxon signed-rank test comparing TabImpute and HyperImpute to address statistical significance: t-test: ($t=5.2051$, $p=1.4 \times 10^{-7}$), Wilcoxon: ($W=92400$, $p=7.6 \times 10^{-7}$). Thus, TabImpute is significantly better than HyperImpute ($p < 0.001$). The TabPFN paper also compared their model against other ones using normalized negative RMSE values.
>
> 2. Our new experimental results illustrate that the standalone TabImpute model (trained only on MCAR patterns using Linear Factor Models) performs strongly without leveraging the nonlinear power of EWF-TabPFN.  The imputation power comes from the linear factor model alone. We attribute this success to the two primary inductive biases we discuss in the top comment.
> 3.  After fixing the normalization error in our training code, TabImpute’s performance on Censoring-MNAR is more in line with other methods. We attribute this performance difference to TabImpute’s nature as a generative model. Generative approaches typically yield the most significant gains on high-missingness patterns by effectively modeling the joint distribution when observed signal is more scarce. On the other hand, patterns like Censoring-MNAR present a lower fraction of missing data. In such settings where we have more information, the relative advantage of a generative model over discriminative methods is less pronounced.
> 4. We updated the manuscript to clarify that our TabImpute model does not use SCMs for synthetic data generation and that we only use linear factor models for generating underlying matrix values. We have added Table 8 to Appendix Section A.3. which displays results with nonlinear factor models.
>
> We hope that we have addressed your feedback sufficiently and will happily provide further comments if you have any remaining questions.

---

> > ### Author Response · Authors · 2025-11-23
> >
> > Hello, and thanks again for the thoughtful reviews. We’ve uploaded our rebuttal and hope it resolves the concerns you raised. Please feel free to let us know if any additional clarification would be helpful.

---

> ### Comment · Reviewer_HSxa · 2025-11-25
> **Major Concerns**
>
> We thank the authors for their responses; however, several new major and minor concerns have arisen, given the current state of the updated manuscript.
>
> ## Major Concerns
>
> **1. Discordant Results with Well-Established Baselines**
>
> The updated manuscript presents several results in Figure 1 that are discordant with established findings in the tabular imputation literature [1-7]. Most concerningly, DiffPuter, ReMasker, and many alternative approaches perform worse than ColMean.
>
> ColMean has historically been used and reported in the field not because it performs well, but because it serves as a useful "negative control." If evaluation indicates that established methods like MICE (which, in the worst case, can still estimate the mean via the regression intercept) perform much worse than ColMean, this signals a fundamental issue with either the evaluation process or the metrics employed. Indeed, if one were to report R^2 for imputation performance as other works have done, ColMean should deterministically equal 0.0 (since correlation is shift- and scale-invariant).
>
> Multiple works over the years have independently replicated that ColMean underperforms ReMasker, DiffPuter and similar methods [1-7]. The work's inability to replicate this "negative control" behavior indicates that either (a) the evaluation is biased and inaccurate, and/or (b) the proposed MissBench itself has fundamental flaws. Either concern is sufficient to mandate complete reprocessing and reanalysis, and calls into question all reported results and claims.
>
> If the issue lies with MissBench rather than the evaluation, this would render the proposed benchmarking dataset unsuitable for the imputation field and could mislead future researchers toward approaches unlikely to generalize to real applications. Additionally, ColMean is only expected to be unbiased under MCAR conditions and should deterministically worsen under MAR and especially MNAR. Given that the majority of this dataset comprises MNAR patterns (regime where ColMean suffers), the finding that ColMean achieves "median-level" performance is particularly concerning.
>
> **2. Missing Reference and Statistical Inconsistencies in Figure 1**
>
> I have been unable to locate any reference to Figure 1 (and also Fig. 3) in the main text. While this alone might constitute a minor oversight easily corrected (though it suggests insufficient rigor in manuscript preparation), a substantial concern emerges from examining the figure itself.
>
> Figure 1(a) claims to report 95% confidence intervals. Visual inspection suggests these CIs are <0.1. However, Table 1 reports standard deviations for TabImpute and HyperImpute are strictly >0.19, which should produce substantially larger 95% CIs. I also believe the repoted numbers are standard error of the mean (SE), defined as the standard deviation of the point estimates across the observed or bootstrapped estimands. (Please correct me if this is not the case.)
>
> Upon closer reading, the caption states that Figure 1 results are "calculated for each method, normalized within a dataset, and averaged across datasets and 13 missingness patterns." While this normalization procedure may partially explain the discrepancy, I am not convinced it fully accounts for the large difference in variance. Unfortunately, the raw results are not reported in the appendix, not allowing verification. Given the concerns raised in Major Concern 1, I cannot extend the benefit of the doubt on this matter.
>
> **3. Substantial Mid-Review Update**
>
> After the initial review, the authors significantly updated the paper with the following statement: "We found an error in our training code where we normalized a training data matrix before inducing missingness... TabImpute achieved state-of-the-art performance..." This update fundamentally alters the original findings and claims of the original submission.
>
> (Note to the AC: Please feel free to strike and disregard this point if you deem this major update appropriate given review guidelines.)

---

> > ### Comment · Reviewer_HSxa · 2025-11-25
> > **Minor Concerns**
> >
> > ## Minor Concerns
> >
> > - The reported paired t-test and Wilcoxon test report p<8x10^-7. What exactly are these tests comparing? Is it two-sided? Regardless, the very large standard deviations reported in most of the tables make it very unlikely to result in such a statistically significant result.
> >
> > - It is still unclear why training on non-linear data makes TabImpute’s performance worst than HyperImpute. Linear models are just a special case of non-linear models. And indeed, some of the MNAR conditions like clustering and latent-factor are modeled under non-linear missingness. See the use of the sigmoid function to generate the mask (in the appendix methods). Furthermore, the finding that model “trained only on MCAR patterns using Linear Factor Models) performs the best” is very surprising, given a majority of the evals are under MNAR. The “two primary inductive biases” are not sufficient to explain this since, even if the model has non-linearities the training data does not. This can indicate an issue with evaluation or bias in the model.
> >
> > - Latent-Factor-MNAR is a special case of MNAR-WS (block) see [8] and [9]. Furthermore, the concerns about sequential missingness patterns are still unaddressed. It was never my claim that the pattern is not realistic. My core issue was that “Such patterns only make sense for repeated measures of the same feature over time, which doesn't appear to be the case here.” This would require there to be a specific relationship (a sequential/temporal one) between a feature in your fully observed X before one even simulates the mask. Unless I missed it, these simulations have not been updated. It's just the text has just been updated to better make a case for MNAR-cencoring (which is certainly an important setting).
> >
> > - CACTI can be run without context and has the same runtime as ReMasker (see [7]). “CACTI requires dataset-specific training” is not a valid reason since all other benchmarked methods need the same. Also, please note that HyperImpute can only perform in-sample imputation. The rest of the methods can be trained on a subset of the dataset and run in inference-only mode on a held-out dataset. It's unclear based on the manuscript if all the reported results are in-sample imputation or reported from a held-out test split.
> >
> > - Claiming SOTA in tabular imputation is a high bar and needs the reporting of other metrics like R^2 and WD. It's clear from the direct reporting of other non-normalized metrics in Table 2 that TabImpute has not reached that bar yet. Reporting common metrics in the imputation literature is important to contextualize the results within the many prior works that exist.
> >
> > Overall, the current version of the manuscript has key issues with analysis and how the results are presented. It is in need of a major revision and not quite ready for publication in its current form.
> >
> > #### References
> >
> > [1] Missing Data Imputation using Optimal Transport (ICML, 2020)
> >
> > [2] not-MIWAE: Deep Generative Modelling with Missing not at Random Data (ICLR, 2021)
> >
> > [3] A Benchmark for Data Imputation Methods (Front. Big Data, 2021)
> >
> > [4] HyperImpute: Generalized Iterative Imputation with Automatic Model Selection (ICML, 2022)
> >
> > [5] ReMasker: Imputing Tabular Data with Masked Autoencoding (ICLR, 2024)
> >
> > [6] DiffPuter: Empowering Diffusion Models for Missing Data Imputation (ICLR, 2025)
> >
> > [7] CACTI: Leveraging Copy Masking and Contextual Information to Improve Tabular Data Imputation (ICML, 2025)
> >
> > [8] A Complete Characterization of Structured Missingness (2023)
> >
> > [9] Inference and Missing Data (Biometrika, 1976)

---

> > > ### Author Response · Authors · 2025-11-26
> > > **Reply to minor concerns**
> > >
> > > On your second point, we were also surprised that the model trained on linear factor models with MCAR missingness performed the best. We tested training with other patterns mixed in and on a nonlinear factor model as well, as discussed in our updated results section. In this work, our investigations and findings were empirically driven, and we plan on determining a theoretical reason for this model’s superior performance in future work.
> > >
> > > While we certainly would like to include CACTI in our evaluations, and were able to get CACTI without context to run for several of our benchmarks, we found that we ran into an internal error with the mask generation on too many of our datasets for us to include CACTI at the moment. If this error could be fixed in the future, then we will definitely include it in our experiments. Similarly, we would also like to add DiffPuter and ReMasker in our main results, but were unable to get their code to work on MissBench, even for simple MCAR masks. We will continue to investigate these code bases to add these new methods to our evaluations.
> > >
> > > We hope that we have alleviated your concerns and look forward to further discussion.

---

> > ### Author Response · Authors · 2025-11-26
> > **Reply to major concerns**
> >
> > Thank you for the comprehensive response. We first address your second point:
> >
> > **2nd Point:** The standard deviation of samples measures its spread around the average, whereas standard error—used for constructing confidence intervals—is a normalized version of the standard deviation by the square root sample size. The confidence interval reported in Figure 1 uses the mean and standard deviation calculated across $42 \times 13$ negative RMSE values (42 datasets and 13 missingness patterns), but the width of the interval requires a division of the standard deviation by the square root of the sample size $42 \times 13$, multiplied by $1.96$ (corresponding to $95\%$ confidence). For consistency, we have revised our manuscript so that tables now contain standard errors, which are the standard deviation divided by the square root of sample size, as used to calculate confidence intervals.
> >
> > The division of the standard deviation by the sample size explains the narrow width of the confidence interval and the large difference you observed. The raw results would require presentation of $42 \times 13$ different performance metrics across several imputation techniques; we decided not to present such a large number of raw results as it does not convey additional value, but rather hurts comprehension. Our benchmark, model weights, and evaluation code is open-sourced, as is our model and every method we tested against. Thus, these results can be replicated by anyone with a standard GPU. Additionally, our method only requires a few lines of code to use on any matrix with missing values since the pre-trained model is automatically downloaded from HuggingFace:
> > ```python
> > from tabimpute.interface import TabImpute
> > import numpy as np
> >
> > imputer = TabImpute(device='cpu') # cuda if available
> > X = np.random.rand(5, 5)
> > X[np.random.rand(*X.shape) < 0.1] = np.nan
> > imputed_X = imputer.impute(X.copy())
> > ```
> >
> > The paired negative RMSE values of TabImpute and HyperImpute for each dataset and missing pattern pair (total of $42\times 13$ pairs) are used for the Wilcoxon and the pairwise t-test, both tests (prior non-parametric and latter parametric) aiming to determine which method has a higher negative RMSE compared to the other. Again, the standard deviations presented in Table 1 do not directly translate to assessing significance within confidence intervals and statistical tests. So the significance observed in our pairwise t-test and Wilcoxon test aligns with the conclusion of the confidence intervals in Figure 1.
> >
> > Lastly, thank you for pointing out the missing references. We have included references in the updated manuscript.
> >
> > **1st Point:** We have updated our manuscript to remove the results for DiffPuter because, on further analysis, their publicly available code was not working on our benchmark. We will examine this further, but we do not wish to misrepresent their work. In general, our benchmark contains datasets that are much smaller than ones tested in several previous papers. For instance, DiffPuter is only tested on datasets with more than 12,000 rows, whereas we test on datasets with fewer than 250 rows. It is possible that the DiffPuter internal optimization for diffusion does not work for small datasets, given that it always outputs a constant value per column for the datasets in MissBench. The experiments in CACTI were also only for much larger datasets, with most datasets having over 10,000 rows, and the smallest dataset having 4,000 rows. The regime tested here, though, requires the models to work with much less data. In general, for neural network-based diffusion, a vast amount of data is required [1, 2].
> >
> > TabImpute, on the other hand, is based on TabPFN, which excels on smaller datasets due to its strong prior. Similarly, ForestDiffusion uses XGBoost for diffusion, which induces a more suitable prior for tabular data. In future work, we plan on investigating this dataset-size hypothesis further by testing our method on larger datasets as well.
> >
> > Finally, we have now added a table in the appendix (Table 7) with $R^2$ values calculated column-wise for each method. Column Mean is at the bottom with a value of 0, and EWF-TabPFN narrowly beats TabImpute in terms of $R^2$. We believe that training the model further and with more complex patterns should improve this performance.
> >
> > We hope that we have addressed your concerns. Please let us know if we can clarify anything further.
> >
> > References:
> > [1] Wang, Zhendong, et al. "Patch diffusion: Faster and more data-efficient training of diffusion models." Advances in neural information processing systems 36 (2023): 72137-72154.
> > [2] Huang, Rui, et al. "Diffusion dataset condensation: Training your diffusion model faster with less data." arXiv preprint arXiv:2507.05914 (2025).

---

### Official Review · Reviewer_uvPA · 2025-10-31

**Soundness:** 2
**Presentation:** 1
**Contribution:** 2
**Rating:** 4
**Confidence:** 4

**Summary:**

The paper introduces TabImpute, a pre-trained Transformer for zero-shot imputation in tabular data, building on TabPFN. It proposes an efficient entry-wise featurization and a synthetic data generation pipeline with realistic missingness patterns. Additionally, it presents MissBench, a benchmark spanning 42 OpenML datasets and 13 missingness scenarios across multiple domains.

**Strengths:**

- The idea of leveraging Prior-Data Fitted Networks (PFNs) for imputation tasks is interesting and opens new directions in amortized inference under missing data.

- The paper centers on the underexplored challenge of MNAR imputation, which is important in real-world tabular applications.

- The use of in-context learning for tabular data continues a promising line of work and may inspire future extensions beyond classification.

**Weaknesses:**

- The methodological contribution is minimal, as the proposed TabImpute appears to be a near-direct adaptation of TabPFN, with only slight modifications to the attention mechanism and task framing.

- The related work section is overstated and omits many relevant deep generative models for tabular imputation, particularly those tailored to MNAR data (e.g., [1–3]). These should be cited and, where possible, used as baselines.

- The paper lacks scientific rigor in some claims, such as criticizing TabPFN's performance on the proposed benchmark without sufficient grounded analysis or discussion of why it fails.

- Key technical details are missing, especially in the Introduction. The exposition prioritizes benchmark specifications over a clear and detailed presentation of the model itself.

- Experimental comparisons are limited to only a few methods and primarily on the authors' own benchmark, making it difficult to assess broader performance. While more baselines appear in Figure 1, there are no implementation details, and surprisingly poor performance is reported for some models without explanation. In fact, some claims about baseline limitations (e.g., lack of GPU support for GAIN, HyperImpute, and MIWAE) are factually incorrect and should be corrected.

**Questions:**

I have no further questions except for the points mentioned above.

---

> ### Author Response · Authors · 2025-11-19
>
> Thank you very much for the thoughtful review. We do not see several papers which you cited, but if you post them, we would be happy to include them in our paper. We hope to address your comments sufficiently below:
>
> > You write: The methodological contribution is minimal, as the proposed TabImpute appears to be a near-direct adaptation of TabPFN, with only slight modifications to the attention mechanism and task framing.
>
> We agree that our adaptation of the TabPFN architecture is slight. However, we do this because we believe that it is a very powerful architecture that does not require much modification to apply to other tabular problems. From a scientific point of view, we seeked to address whether the training procedure alone could be enough to create a better imputation model, which we have shown in our top-level comment with our new TabImpute model that performs the best without any ensembling with EWF-TabPFN.
>
> > You write: The paper lacks scientific rigor in some claims, such as criticizing TabPFN's performance on the proposed benchmark without sufficient grounded analysis or discussion of why it fails.
>
> While we would definitely like to know why applying TabPFN column-by-column does not work well for data imputation, since TabPFN is a black-box model, we are unable to determine which part of its training procedure (which is not open-source) leads to this poor performance. However, we show through EWF-TabPFN’s much better performance that the main source of error lies in how the model is used, not the model itself.
>
> > You write: Key technical details are missing, especially in the Introduction. The exposition prioritizes benchmark specifications over a clear and detailed presentation of the model itself.
>
> Since we do not modify the architecture of TabPFN that much, and due to space constraints, we did not wish to repeat an explanation of TabPFN’s architecture. We agree that if there was more space, it would be clearer to include more of an explanation of the architecture.
>
> > You write: Experimental comparisons are limited to only a few methods and primarily on the authors' own benchmark, making it difficult to assess broader performance. While more baselines appear in Figure 1, there are no implementation details, and surprisingly poor performance is reported for some models without explanation. In fact, some claims about baseline limitations (e.g., lack of GPU support for GAIN, HyperImpute, and MIWAE) are factually incorrect and should be corrected.
>
> We open-source our model code, parameters, training code, and the benchmark as well. We conjecture that many other non-generative methods rely too heavily on observed data and thus perform poorly when missingness is higher either globally or locally (within a row or column), which is the case in our benchmark, especially for structured missingness patterns. Note that this behavior would not have been observed in the previous papers because they only tested on the simplest forms of missingness, MCAR and specific MAR patterns. For instance, MissForest tests only on MCAR data at 10, 20, and 30% missing. MIWAE tests using MCAR data with 50% missingness. HyperImpute tests under MCAR, Col-MAR, and Self-Masking-MNAR all at 30% missingness. Each of these papers use datasets from UCI, which is a subset of OpenML, the repository of public datasets that we use. Note that we could not use the exact same datasets as these papers because many of these datasets have categorical variables, which we do not yet support.
>
> **GPU Support for Other Methods:** Thank you for pointing out that other methods also are GPU-compatible. We agree that HyperImpute, GAIN, and MIWAE support GPUs. In fact, the HyperImpute implementation of these three methods automatically uses a GPU if it is available, which was always true for our experiments. We have updated our runtime plot to reflect that these methods used a GPU.
>
> We wish to thank you again for your thoughtful review. We hope that we have addressed your concerns and look forward to discussing further.

---

> > ### Author Response · Authors · 2025-11-23
> >
> > Hello, and thanks again for the thoughtful reviews. We’ve uploaded our rebuttal and hope it resolves the concerns you raised. Please feel free to let us know if any additional clarification would be helpful.

---

### Official Review · Reviewer_EfQp · 2025-11-03

**Soundness:** 2
**Presentation:** 2
**Contribution:** 2
**Rating:** 2
**Confidence:** 4

**Summary:**

TabPFN which is an pretrained model for tabular data prediction. This work seeks to do the same but for imputation. They build upon TabFPN, but propose a faster featurization method with a synthetic data pipeline with different missingness patterns. They show much better performance on many metrics.

Figures 1 and 2 are great, it shows its better than even MissForest, but runtime stays low. Hmm the only thing I dislike is that the y-axis on Figure 1b is log-linear which really distort the actual gap between methods.
The figures are really nice.

It would be great to spend a bit more time explaining the TapPFN baseline arch and data setup.

Entry-wise featurization is a bit of a fancy word to say (i, j, X[i,:], X[:,j]). I'm not sure I understand. (i, j, X[i,:], X[:,j]) has shape 2+n+m and there are m x n of it, so the shape is [2+n+m,m,n]. How is it a matrix? And how is the target Xij*? Because this means the input is much bigger than the output. Can you clarify and maybe explain it a simpler way.

" We use TabPFN’s base architecture with one modification: removing the attention mask to allow training points to attend to test points. " This seems very wrong. I don't understand how you can justify this. You can impute train data based on train+test data, there is obvious leakage of the test set. Am I missing something here? Because this seems very wrong.

Its interesting that training on simpler data generation process worked better, this may be because of simplicity inductive bias.

Of course sequential training is worse, it make sense. Its nice to include remarks like this in the paper.

The ensembling make sense, but can you add a bit of details. Like having a figure showing inference of your method would be great.

TabImpute struggle with many missingness types. I get that TabImpute+ might be better with only the 3 types of missiness, but is it true also for TabImpute? Because it seems to struggle massively at Censoring-MNAR, Polarization-MNAR, Soft-Polarization-MNAR, Panel-MNAR, Col-MAR, Self-Masking-MNAR. In fact how to explain poor performance on Self-Masking-MNAR if its trained on that??? And what do you mean by zero-shot method, its not defined as far as I can tell. Idk why the numbers are different in Table 2, please explain.

Evaluated on 42 datasets is great but they are pretty small. I assume its a limited of the TabPFN line of work since you need to attend through the whole dataset, so thats fine.

"(ii) enhancing our method to support categorical data" What do you mean, doesn't your method super categorical data? Are you only using one-hot features?


Overall I really like the direction, its interesting and powerful. My big concern is the inclusion of the test set in the imputation self-attention. I really dont't understand how this can be justified. And is it only applied during training or also at evaluation. We need more details on this, does it mean that test points can attend to train post, because that is clearly wrong if thats the case. Training points should be used to produce the imputation of training and test points, test points should be used to produce test point, but test points CANNOT be used to produce train points.

The only metric is this normalized RMSE, but what about other metrics. Currently it says nothing about diversity of imputations. This is actually quite important in the context of multiple imputations where Rubin showed that you need diversity to get good estimates. And diversity I assume would be affected by your random shuffling and whatnot at inference, it would be good to see. Its okay if there is no diversity, then it become a better MissForest, but we need to know if its diverse or not because Multiple Imputation folks will not use it if its not diverse. I recommend using some of the metrics in Table 3 of https://arxiv.org/abs/2309.09968.

There are many things that are justified without numbers, it would be great to see an ablation, this would explain why use only the 3 missing patterns instead of all patterns and the removal of the attention mask. Actually, I see that there is some ablation in the appendix, but its missing the removal of the attention mask and it would be great to have more context and explanation about the ablation results.

Right now, I cannot give a good score giving my concern on the self-attention mask, but if its resolved (maybe I understood wrong?), the score would go up. And if the paper was improved a bit (more metrics, especially with diversity in mind, maybe some inference time figure, some cleanup of the text), I would likely increase it even more since its an interesting method.

**Strengths:**

See "Summary"

**Weaknesses:**

See "Summary"

**Questions:**

See "Summary"

---

> ### Author Response · Authors · 2025-11-19
>
> Thank you for the comprehensive and thoughtful comments. We are very pleased to see that you appreciate our proposed method and are grateful for your suggestions on improving the manuscript. We hope to address each of your suggestions and comments both below and in our top-level comment. To make navigation easy, we point to where each of your comments is addressed:
> - Attention mask removal and featurization: Below
> - Improving TabImpute’s imputation accuracy: Top-level comment
> - Supporting categorial features: Top-level comment
> - Adding multiple imputation metrics: Top-level comment
> - Comparison with another DGP for training: Top-level comment
>
> **Attention mask and featurization:** We agree that we should have taken more care with our explanation of the attention mask modification. To explain this better, we first need to give a brief explanation of how TabPFN works: the model acts on entire matrices of data. The model weights are pre-trained on a massive corpus of synthetically-generated matrices. The model is then applied with fixed weights on new real-world matrices for evaluation. These models are called zero-shot because the model weights are fixed after pre-training and are not updated on real-world data.
>
> We agree that there can be confusion with our usage of the words train and test. When we are discussing a matrix set up for supervised learning, with features along the columns and samples along the rows, we refer to the rows with labels as train rows and the rows without labels as test rows. In TabPFN literature, the train rows are referred to as input rows, and test rows are referred to as query rows. (Note that query rows do not have any labels, so there is no data leakage.) These notions are different from train and test datasets/matrices. For our missingness tasks, the train matrices are synthetically generated and the test matrices are the ones in MissBench. The model is never trained on any data from MissBench. To remove any future confusion about this terminology, we updated the manuscript to go back to using input and query rows.
>
> We demonstrate our featurization and attention on a small matrix with missing values:
> $$\\begin{pmatrix}
> 	1 & NaN & 3 \\\\
> 	NaN & 5 & 8 \\\\
> 	2 & 4 & NaN
> \\end{pmatrix}$$
> We featurize this matrix using our entry-wise featurization approach:
> $$\\begin{pmatrix}
> 	1 & 1 & 1 & NaN & 3 & 1 & NaN & 2 \\\\
> 	1 & 2 & 1 & NaN & 3 & NaN & 5 & 4 \\\\
> 	1 & 3 & 1 & NaN & 3 & 3 & 8 & NaN \\\\
> 	2 & 1 & NaN & 5 & 8 & 1 & NaN & 2 \\\\
> 	\vdots & \vdots & \vdots & \vdots & \vdots & \vdots & \vdots & \vdots  \\\\
> 	3 & 3 & 2 & 4 & NaN & 3 & 8 & NaN
> \\end{pmatrix}$$
> The target vector, corresponding to the rows, are the values of each cell (whether labeled or un-labeled): $y = (1, NaN, 3, NaN, 5, 8, 2, 4, NaN)$. We have 6 labeled entries and 3 unlabeled entries. $X$ and $y$ are inputted into the model and attention is performed over all of the rows and columns. During pre-training, we have access to the true values of the NaN’s in the y vector. However, these values are only used to update the weights of the model via the loss function. The true values are never inputted into the model. The model internally handles NaN inputs by replacing their values with the column mean in the input matrix and then appending a binary matrix indicating which values are NaN. Internally, the NaN values in the y input are replaced with the mean of the non-NaN values (observed values) in the y vector.
>
> In TabPFN’s case, the authors did not want any query points to attend to other query points and did not want any input points to attend to query points. The reason for this is because they wanted the internal fitted function of TabPFN to not rely on query points, just as classical linear regression models rely only on input points to fit weights and do not change when applied to different query points. Note, however, that this choice of attention mask is merely to enforce a heuristic of what should guide the internal fitted function of TabPFN.
>
> For TabImpute, the query rows are the rows that are unlabeled because their respective values in the original matrix are missing. We do not need the attention mask because we do not have a prior on what kind of internal function the transformer should use.
>
> Another type of transformer that does not always have an attention mask is a vision transformer (ViT). The reason for this is because any part of an image can attend to any other part. There is no inherent ordering that must be enforced.
>
> We hope that we have addressed your concerns here. Please let us know if there is anything else that we can clarify or add to our manuscript.

---

> > ### Comment · Reviewer_EfQp · 2025-11-20
> >
> > Thanks for the explanation that the test rows don't have label, this clarify things. And this is leak-free, I agree. A few models trained for ARC-AGI do the same thing.
> >
> > The fact that now you don't need to ensemble to perform best is great. You added many ablations and metrics as asked. I think that this is a strong paper. The only major limitation right now is the fact that it doesn't work with categorical data since most tabular data include mixed-type data.
> >
> > I will update my score accordingly.

---

> ### Author Response · Authors · 2025-11-24
> **Added support for categorical features**
>
> Thank you. We have now added support support for categorical variables via one-hot encoding: We encode categorical variables, impute the missing 0/1 values in this space, and then run softmax over of the imputed numerical values in the one-hot encoded columns. This outputs probabilities over the classes within a categorical column. For the final imputed category, we choose the class with the highest probability. We test this method on 53 datasets from OpenML in Table 4 in the section A.1 in the Appendix and find that we have competative performance in terms of ROC AUC with HyperImpute and MissForest. We believe that including categorical features in the training process or using an optional categorical output head in the network will improve performance and plan on testing these in the future.

---

### Author Response · Authors · 2025-11-19
**New results and common comments**

We thank each reviewer for their careful analysis and helpful recommendations. We have used these recommendations to greatly improve the paper:
- We found an error in our training code where we normalized a training data matrix before inducing missingness. We trained a new model and its performance on complex patterns improved drastically: TabImpute achieved state-of-the-art performance, and this time using only our trained model, not an ensemble with EWF-TabPFN, as shown in Table 1 in our updated manuscript. We no longer see the poor performance of our model on Censoring-MNAR.
- Added multiple imputation metrics: median absolute deviation, min RMSE, and avg RMSE to demonstrate that TabImpute can be used easily in multiple imputation settings. We discuss this in our results section and Table 2 in the updated manuscript.
- Added ForestDiffusion, shown in Fig. 1 of the updated manuscript and full RMSE table in the appendix. We were unable to get ReMasker or Diffputer working on MissBench.
- Added additional citations for each MNAR pattern in the appendix to show each fits into previous literature.

**Computational Complexity:** TabPFNv2 details their model computational requirements on a table with $n$ samples and $m$ features to be $O(n^2 + m^2)$ due to their $1$-dimensional attention components over each axis. With our entry-wise featurization we build a table of $nm$ samples and $n+m+2$ features, i.e., one for each entry, the corresponding column, row, and the position, giving a complexity of $O(n^2m^2)$. Note that in most settings $m$ is small and the quadratic complexity in sequence length is a known pitfall of attention based mechanisms. We used the TabPFN architecture to first evaluate the question of whether this architecture could be state of the art on imputation techniques. Some directions we leave for future work are:
- Local or block-sparse attention limits each token’s neighborhood to nearby cells.
- Linear or low-rank attention replaces the quadratic attention matrix with kernel feature maps, so attention scales almost linearly in the number of tokens.

**Categorical Columns:** The synthetic training data didn't contain categorical variables. There are two ways TabImpute can support categorical variables in the future: (i) After encoding categorical variables as one-hot features, we can apply the same featurization technique of TabImpute, (ii) when the target variable is an ordered categorical variable, we can discretize the numeric output. A new synthetic data-generating process (DGP) that includes categorical variables and a head that allows categorical outputs is necessary for a reasonable model that is expected to do well in imputations for matrices including categorical data. For training and inference, there is no immediate bottleneck we observe. We tested one-hot encoding in Table 4 in the Appendix.

**Linear Factor Model for Synthetic Training Data:** Our new results show that TabImpute (trained only on MCAR patterns using Linear Factor Models) performs the best, no longer requiring the nonlinear power of EWF-TabPFN. We attribute this success to two primary inductive biases:
- Low-Rank Approximation: Real-world tabular data can be universally approximated by low-rank linear structure (Udell and Townsend, https://arxiv.org/pdf/1705.07474).
- Conditional Distribution Modeling: The objective is to learn the conditional distribution of missing entries given observed features. Our model approximates this via the Transformer architecture (with nonlinear activation functions), even when the underlying DGP is linear.

We trained a model on a nonlinear FM and it performed slightly worse than the LFM-trained model, presented in Table 8 in our updated appendix.

**Citation and Motivations for Missing Data:** Sequential policies are one of the most widely adopted tools to quantify causal effects, and counterfactual learning is conveniently expressed as a matrix completion problem [2]. Seq-MNAR exemplifies these sequential policies into missingness patterns, and TabImpute provides a simple solution of predicting the missing counterfactuals. We provide a direct connection of our DGP with the taxonomy provided in [1], which claims an extension to that of Rubin. We further discuss how our missing patterns (especially MNAR) extend upon the framework of [1]. Note that our MCAR pattern exactly matches the Definition 1 of [1] and the MAR pattern also matches the Definition 2 of [1]. For the 11 MNAR patterns, as we have direct access to the full underlying matrix values, we not only cover how propensities depend on the observed and unobserved entries $X_{obs}$ and $X_{missing}$ (as specified in the Definition 3 of [1]), but also on the underlying latent structure of the matrix X.

We look forward to further discussing our new results and addressing any more questions.

Citations:
[1] A Complete Characterization of Structured Missingness (2023)

[2] Athey et. al., 2021, https://arxiv.org/abs/1710.10251

---

### Meta-Review · Area_Chair_W7Ya · 2026-01-12

**Summary:**

The paper proposes a foundational model for zero-shot imputation of missing values in tabular data, named TabImpute and extending TabPFN, and a new benchmark called MissBench. Reviewers found the direction interesting and promising, including the focus on data with MNAR values, and, for some reviewers, strong empirical performance. However, reviewers raised significant concerns about the baseline coverage (and therefore unsupported claim about SOTA performance), lack of scientific rigour in claims, insufficient grounding for Seq-MNAR in tabular data, lack of support for categorical data, minimal methodological contributions, omission of relevant related models in literature review, limitation in computational complexity and scalability, leakage in attention mask, discordant results with well-established baselines, and that the paper was substantially updated mid-review since the authors found a bug in their training code.

**Reviewer Concerns:**

The rebuttal addressed concerns about leakage in the attention mask, added a workaround for categorical data, and offered a motivation for the missingness mask. However, significant issues remained regarding discordant results with well-established baselines (which raised concerns about the benchmark's validity), the mid-review update, the method's scalability, and the continued issue of support for categorical data (for some reviewers).

**Reviewer Scores:**

Reviewers EfQp would like to increase their score, as the concern about attention mask leakage was addressed (and the reviewer stated so). The scores of reviewers uvPA and V9f3 would likely remain unchanged (uvPA raised concerns about minimal methodological novelty, which were not addressed, and V9f3 directly stated so). Reviewer HSxa would likely decrease their score, as the reviewer raised major concerns about the mid-review update and the discordant results in it. Given that the overall scores would approximately remain unchanged (with one review increasing and another decreasing scores), and that major changes were made mid-reviewer, the paper needs a new full round of reviews. Therefore, I recommend rejecting the paper and encourage the authors to send a majorly revised paper to the next machine learning conference.

---

### Decision · Program_Chairs · 2026-01-26

Reject